# SimpleRL-Zoo: Investigating and Taming Zero Reinforcement Learning for Open Base Models in the Wild

**Weihao Zeng**[*1]    **Yuzhen Huang**[*1]    **Qian Liu**[*2]    **Wei Liu**[1]    **Keqing He**[3]
**Zejun Ma**[2]    **Junxian He**[1]
[1]HKUST    [2]TikTok    [3]Meituan
https://github.com/hkust-nlp/simpleRL-reason

## Abstract

DeepSeek-R1 has shown that long chain-of-thought (CoT) reasoning can naturally emerge through a simple reinforcement learning (RL) framework with rule-based rewards, where the training may directly start from the base models—a paradigm referred to as *zero RL training*. Most recent efforts to reproduce zero RL training have primarily focused on the Qwen2.5 model series, which may not be representative as we find the base models already exhibit strong instruction-following and self-reflection abilities. In this work, we investigate zero RL training across 10 diverse base models, spanning different families and sizes including LLama3-8B, Mistral-7B/24B, DeepSeek-Math-7B, Qwen2.5-math-7B, and all Qwen2.5 models from 0.5B to 32B. Leveraging several key design strategies—such as adjusting format reward and controlling query difficulty—we achieve substantial improvements in both reasoning accuracy and response length across most settings. However, by carefully monitoring the training dynamics, we observe that different base models exhibit distinct patterns during training. For instance, the increased response length does not always correlate with the emergence of certain cognitive behaviors such as verification (i.e., the "aha moment"). Notably, we observe the "aha moment" for the first time in small models not from the Qwen family. We share the key designs that enable successful zero RL training, along with our findings and practices. To facilitate further research, we open-source the code, models, and analysis tools.

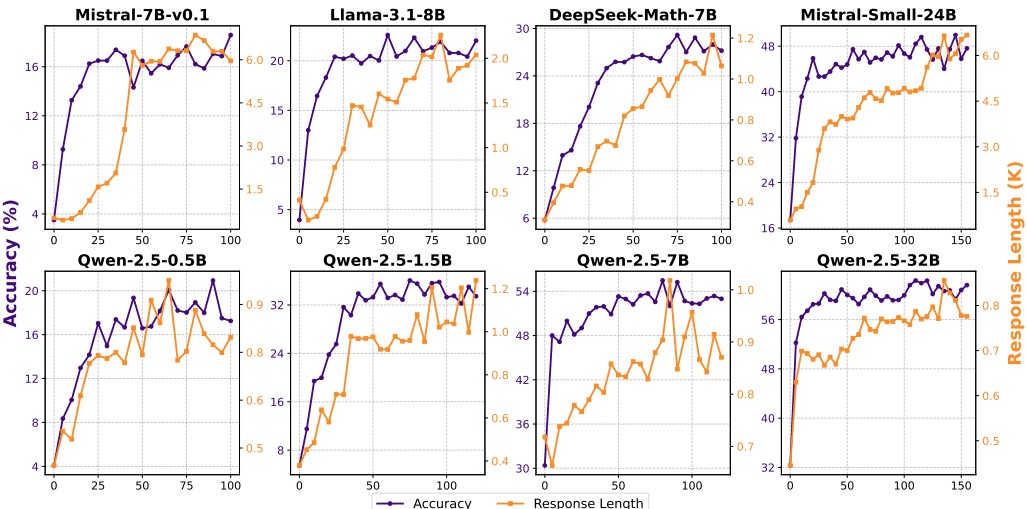

Figure 1: Accuracy and response length across training iterations for different models, averaged on GSM8K, MATH500, Minerva Math, OlympiadBench, AIME24, and AMC23. Per-benchmark results are in Figure 11 (Appendix D). All training starts from base models.

*Equal Contribution. Correspondence to Weihao Zeng (wzengak@connect.ust.hk), Yuzhen Huang (yhuanghj@cse.ust.hk), and Junxian He (junxianh@cse.ust.hk).

# 1 Introduction

Large reasoning models, including OpenAI-o1 (Jaech et al., 2024), DeepSeek-R (DeepSeek-AI et al., 2025a), and Kimi-k1.5 (Team et al., 2025), demonstrate remarkable abilities. These models excel at generating long Chains-of-Thought (CoT) (Wei et al., 2022) responses when solving complex tasks and exhibit advanced, reflection-like reasoning behaviors. Recently, DeepSeek-R1 (DeepSeek-AI et al., 2025a) has revealed that starting from pretrained models (i.e., base models), pure reinforcement learning (RL) with rule-based reward can lead to the spontaneous emergence of long CoT and self-reflection behaviors, called the "aha moment". This RL training paradigm starting from base models is often referred to as *zero RL training*.

While the success of zero RL training was initially demonstrated using DeepSeek-V3 (DeepSeek-AI et al., 2025b), a model with 671B parameters, it remained unclear whether such emergent phenomena persist in generally smaller and less capable open base models. Recent open-source efforts exploring zero-training approaches have predominantly centered on the Qwen2.5-series models (Zeng et al., 2025a; Yeo et al., 2025; Xie et al., 2025; Hu et al., 2025; Yu et al., 2025), which, even as base models, exhibit strong instruction-following capabilities and display notable cognitive behaviors such as backtracking and verification from the beginning, as we will detail in §2.4. Moreover, the analyses of model behavior in these studies remain largely superficial, focusing primarily on metrics such as response length and accuracy. These observations neither clearly establish whether the models' reasoning behaviors actually change nor clarify the mechanisms underlying the emergence of effective reasoning, leaving a significant gap in understanding.

To provide a more transparent understanding of zero RL training across different base models in the wild, this paper addresses the following key questions: (1) How do reasoning capabilities develop across various models during zero RL training? (2) Does an "aha moment" still occur for base models that initially lack strong instruction-following and self-verification abilities? (3) What are the critical factors for ensuring successful zero RL training across diverse base models?

To this end, we perform zero RL training across a diverse range of model series and sizes, including Mistral-7B (Jiang et al., 2023), Mistral-24B (Mistral AI, 2025), Llama3-8B (Dubey et al., 2024), DeepSeek-Math-7B (Shao et al., 2024), Qwen2.5-0.5B/1.5B/7B/14B/32B (Yang et al., 2024a), as well as Qwen2.5-Math-7B (Yang et al., 2024b). To maintain simplicity in the training recipe, our experiments rely exclusively on the training sets of GSM8K (Cobbe et al., 2021) and MATH (Hendrycks et al., 2021) datasets for rule-based reward modeling. It is worth noting that we adopt the same training hyperparameters to train all the models. Using GRPO (Shao et al., 2024) as the RL algorithm, combined with several critical factors that we identified, we obtain significant improvements in model accuracy across all base models, along with a notable increase in response length for 9 out of the 10 models, with the exception of Qwen2.5-Math-7B. However, through careful monitoring of training dynamics and reasoning behaviors, we find that different base models exhibit distinct patterns during training. Also, certain specific factors require careful attention to ensure successful zero RL training. Below, we summarize our key findings.

1. Increased response length does not always correspond to an "aha moment" – Interestingly, for most Qwen2.5 models, which form the foundation of most recent open-source efforts, we do not observe a rise in the frequency of certain cognitive behaviors, such as self-reflection, despite the increase in response length. (§2.4)

2. For the first time, we observe a significant increase in the frequency of specific cognitive reasoning behaviors, such as verification, in small models outside the Qwen family, notably in the Llama3-8B and DeepSeek-Math-7B models. (§2.4)

3. Enforcing rigid format reward (e.g., enclosing answers within boxes) (DeepSeek-AI et al., 2025a) significantly penalizes exploration (Singh et al., 2023; Wang et al., 2024), particularly for base models that initially struggle with instruction following. This restriction lowers their performance ceiling and often induces overthinking behaviors (Chen et al., 2024). (§3.1)

4. The difficulty level of the training data must align closely with the base model's intrinsic exploration capabilities, otherwise zero RL will fail. (§3.2)

5. In contrast to the observation in Shao et al. (2024), zero RL training lifts pass@k accuracy by 10-30 absolute points, a strong evidence confirming zero RL training is not just reranking responses. (§2.3)

6. We revisit the traditional training pipeline that performs SFT to learn to follow instructions before RL training. Specifically, we use conventional SFT datasets as a cold start for RL—a de facto approach prior to the release of DeepSeek-R1. While high-quality CoT data (Li et al., 2024) can rapidly enhance a base model's performance through imitation, we find that it significantly limits the model's ability to explore freely during RL. This constraint diminishes post-RL performance and suppresses the emergence of advanced reasoning capabilities. (§4)

## 2  On Emerging Reasoning in Zero RL Training

Existing research on zero RL training primarily focuses on Qwen2.5-series models, tracking only superficial metrics like accuracy and response length (Zeng et al., 2025a; Hu et al., 2025; Yu et al., 2025). However, Qwen2.5 models, due to their extensive use of synthetic data during pretraining, already exhibit instruction-following abilities and reflective behaviors, which may not represent base models in diverse scenarios. Additionally, an increase in response length does not necessarily indicate the emergence of cognitive behaviors and can sometimes result from meaningless repetition. To address these issues, this section explores zero RL training across various base models of different sizes and families. By monitoring a range of metrics beyond accuracy and response length, we aim to provide a more comprehensive and transparent understanding of zero RL training for open base models in the wild.

### 2.1  Experimental Setup

**Training Algorithm:**  In our study, we follow the zero RL training recipe in DeepSeek-AI et al. (2025a) using various open base models, employing the GRPO algorithm (Shao et al., 2024). Here, zero RL training refers to RL directly from the base model without any prior supervised fine-tuning (SFT). A detailed introduction to GRPO is provided in Appendix A.

**Training Dataset:**  We use the GSM8K (Cobbe et al., 2021) and MATH (Hendrycks et al., 2021) training datasets. In our experiments, we find that data difficulty is critical for successful zero RL (§3.2) and it is necessary to use data that aligns with the model's capability. To investigate this phenomenon, we categorize the data into three difficulty levels: Easy (GSM8K and MATH lv.1), Medium (MATH lv.1–4), and Hard (MATH lv.3–5), with each category containing roughly 8,000 problems.

**Reward:**  We use a rule-based reward function that assigns +1 for correct answers and 0 for incorrect ones. Unlike prior works (Luo et al., 2025; Chen et al., 2025), we avoid format-based reward, which may hinder exploration, particularly for base models struggling with format adherence, as detailed in §3.1.

**Models:**  We conduct zero RL training experiments on Llama-3.1-8B, DeepSeek-Math-7B, Mistral-v0.1-7B, Mistral-Small-24b-Base-2501, and Qwen-2.5 (0.5B, 1.5B, 7B, 14B, 32B). As we perform experiments for a variety of models, under extremely simple settings with small, simple datasets and only correctness reward, we refer to our obtained models as *SimpleRL-Zoo* to represent a simple training recipe for a zoo of open base models. In our preliminary experiments, we observe that using complex prompts with models that have weak instruction-following capabilities often results in instability during training. Therefore, we apply simpler prompts to some models (Llama-3.1-8B, Mistral-v0.1-7B, and Qwen-2.5-0.5B/1.5B). Examples of these prompts are shown in Figure 10 in the Appendix.

| Model | GSM8K | MATH 500 | Minerva Math | Olympiad Bench | AIME24 (Pass@1) | AIME24 (Avg@32) | AMC23 | Avg. |
|---|---|---|---|---|---|---|---|---|
| *Llama, DeepSeek and Mistral Models* | | | | | | | | |
| Mistral-v0.1-7B | 21.2 | 4.2 | 4.0 | 2.4 | 0.0 | 0.0 | 0.0 | 5.3 |
| ↪ + SimpleRL-Zoo | 75.0 | 15.8 | 6.6 | 4.1 | 0.0 | 0.2 | 10.0 | 18.6 |
| Llama-3.1-8B | 39.7 | 13.6 | 4.8 | 3.1 | 0.0 | 0.2 | 2.5 | 10.6 |
| ↪ + SimpleRL-Zoo | 79.2 | 23.0 | 9.6 | 5.3 | 0.0 | 0.2 | 15.0 | 22.0 |
| DeepSeek-Math-7B | 28.4 | 19.4 | 5.5 | 4.7 | 0.0 | 0.0 | 10.0 | 11.3 |
| ↪ + SimpleRL-Zoo | 78.5 | 39.6 | 21.0 | 12.6 | 3.3 | 0.6 | 20.0 | 29.2 |
| Mistral-Small-24B | 78.6 | 43.6 | 10.7 | 11.6 | 3.3 | 0.5 | 17.5 | 27.6 |
| ↪ + SimpleRL-Zoo | 92.0 | 70.6 | 36.8 | 36.6 | 16.7 | 13.1 | 45.0 | 49.6 |
| *Qwen Series Models* | | | | | | | | |
| Qwen-2.5-0.5B | 36.7 | 15.8 | 4.8 | 2.8 | 0.0 | 0.3 | 12.5 | 12.1 |
| ↪ + SimpleRL-Zoo | 49.5 | 34.4 | 10.3 | 8.9 | 0.0 | 0.7 | 22.5 | 20.9 |
| Qwen-2.5-1.5B | 55.7 | 29.6 | 6.6 | 6.5 | 0.0 | 0.1 | 12.5 | 18.5 |
| ↪ + SimpleRL-Zoo | 74.4 | 59.0 | 20.2 | 21.0 | 6.7 | 4.2 | 35.0 | 36.1 |
| Qwen-2.5-7B | 88.2 | 64.6 | 25.7 | 30.1 | 3.3 | 0.3 | 30.0 | 40.3 |
| ↪ + SimpleRL-Zoo | 91.7 | 78.2 | 38.6 | 40.4 | 20.0 | 15.6 | 62.5 | 55.2 |
| Qwen-2.5-Math-7B | 65.5 | 63.6 | 12.5 | 25.8 | 13.3 | 8.6 | 42.5 | 37.2 |
| ↪ + SimpleRL-Zoo | 90.2 | 80.2 | 37.5 | 39.0 | 40.0 | 24.0 | 70.0 | 59.5 |
| Qwen-2.5-14B | 91.6 | 65.4 | 24.3 | 33.5 | 6.7 | 3.4 | 37.5 | 43.2 |
| ↪ + SimpleRL-Zoo | 94.4 | 80.2 | 40.4 | 44.9 | 23.3 | 14.2 | 57.6 | 56.8 |
| Qwen-2.5-32B | 92.9 | 68.6 | 27.9 | 31.1 | 10.0 | 4.5 | 45.0 | 45.9 |
| ↪ + SimpleRL-Zoo | 95.9 | 82.4 | 42.6 | 46.4 | 36.7 | 27.2 | 67.5 | 61.9 |

Table 1: Detailed performance of various models across multiple benchmarks. The blue lines represent the models trained with our recipe. AIME is evaluated in two ways: Pass@1 (single run) and Avg@32 (average score from 32 runs). For AIME24 (Pass@1) and other benchmarks, baselines use greedy decoding, and models with SimpleRL-Zoo use temperature=1.0 and top-p=0.95. For AIME24 (Avg@32), we sample 32 responses per model with the same settings. Average scores are based on AIME (Pass@1) and other benchmarks.

**Benchmark:** We evaluate performance on standard mathematical reasoning benchmarks, including GSM8K (Cobbe et al., 2021), MATH 500 (Hendrycks et al., 2021), Minerva Math (Lewkowycz et al., 2022), and OlympiadBench (He et al., 2024), as well as on competition-level benchmarks such as AIME 2024 and AMC 2023.

For more experimental setup details, please refer to Appendix B.

## 2.2 Evaluation Metrics

During training, we monitor standard metrics such as accuracy and response length across benchmarks. In our preliminary experiment, we observe that response length as a metric is quite superficial and cannot accurately reflect changes in the model's reasoning behavior. Therefore, we adopt the following metrics additionally:

**Reasoning Behavior Ratio:** To better understand the model's reasoning patterns throughout the training process, we adopt the cognitive behavior framework proposed by Gandhi et al. (2025) and use GPT-4o (Hurst et al., 2024) to identify reasoning-related behaviors, including "Backtracking", "Verification", "Subgoal Setting", and "Enumeration". We compare the consistency between GPT-4o and human annotations of reasoning-related behaviors in the Appendix E. We report the ratio of responses that contain such cognitive behaviors. While some recent studies suggest tracking reflection behavior using related keywords (Yeo et al., 2025; Xie et al., 2025) as monitoring signals, we argue that these keywords only exhibit only a weak correlation with high-level reasoning patterns like reflection and verification. As a result, they fail to adequately capture the development of these reasoning processes. We place the setting details, comparisons of different tracking methods, and reasoning behavior cases of different models in Appendix I.

**Clip Ratio:** In the early stages of training, the base model exhibits weak instruction-following ability and often fails to stop appropriately, resulting in irrelevant or excessively long outputs. After training collapses, the model may also generate repetitive or overly extended responses. Since the model has a fixed maximum context length, such outputs

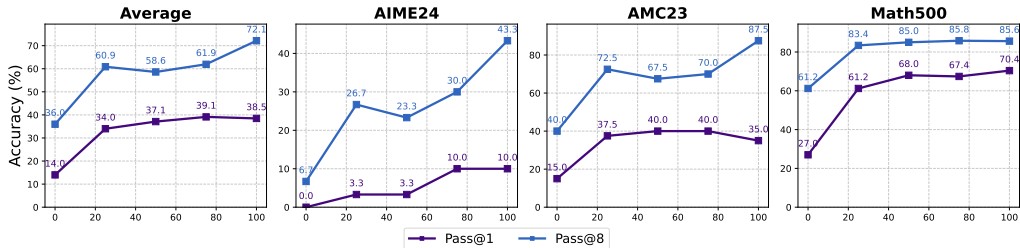

Figure 2: Pass@1 and Pass@8 accuracy over the training iterations of Mistral-Small-24B. The model is trained on the hard data (MATH levels 3–5) as described in §2.1. We evaluate its performance on three benchmarks: AIME24, AMC23, and Math500. The reported average score is the mean across these three benchmarks.

may be truncated during both training and evaluation. To monitor this issue, we define the proportion of truncated outputs as the "Clip Ratio".

**Average Stopped Length:** Generations that are truncated often result from issues such as repetitive patterns or incomplete reasoning, which typically do not contribute to effective trajectories. To account for this factor, we introduce a new metric to track the average length of responses that are stopped under normal conditions.

For more evaluation metrics details, please refer to Appendix C.

## 2.3 Main Results

**Zero RL Training Improves both Accuracy and Response Length Significantly:** Figure 1 and Figure 11 in Appendix D illustrate a steady improvement in both response length and average accuracy across various benchmarks. Table 1 provides a detailed breakdown of the results. Despite using only 8K training samples, we observe significant performance gains for all models. The improvements cover competition-level tests like AIME 2024 and AMC 2023 for most cases. This demonstrates the remarkable generalization capabilities of zero RL training, enabling the model to effectively progress from easier to more challenging problems. In addition to the Qwen series models, we also significantly improve both performance and response length for other models that initially starts with low baselines. For instance, after just 80 training iterations, the DeepSeek-Math-7B's performance increases more than threefold, while its response length grows from around 300 to over 1200 tokens.

**Zero RL Training also Demonstrates Strong Generalization Performance.** We also evaluate the generalization ability of zero RL training using three benchmarks: IFEVAL (Zhou et al., 2023), MMLU (Hendrycks et al., 2020), and GPQA-Diamond (Rein et al., 2024). IFEVAL measures instruction-following capability, MMLU assesses the model's mastery of general knowledge, and GPQA-Diamond is a challenging benchmark that tests domain-specific expertise in chemistry, physics, and biology. Table 2 presents the changes in model performance on IFEval, MMLU, and GPQA-Diamond before and after training. Despite zero RL training being conducted on only 8K math reasoning-related examples, the model generalizes effectively across a range of tasks. Notably, it shows significant gains in instruction-following and general knowledge on IFEval and MMLU, as well as substantial improvements on the challenging GPQA-Diamond benchmark, which spans chemistry, physics, and biology.

**Steady Improvement of Pass@k Accuracy:** As shown in Figure 2, Mistral-Small-24B exhibits robust growth in pass@8. Furthermore, as training progresses, the model's pass@1 results eventually surpass the initial pass@8 results of the base model. By iteration 100, the two metrics differ by more than 30 absolute points on average. This suggests significant potential for further improvements in RL, as our training rolls out 8 responses for each query and pass@8 represents the model's ability to explore correct responses. Surprisingly, the gap between pass@1 and pass@8 does not diminish during training; instead, it widens as training progresses. Figure 3 shows that a significant gap in pass@k persists between the base model and the model after RL training, even at higher values of k – the gap is from 13

| Model | IFEVAL strict-prompt | MMLU Stem | MMLU | GPQA | Avg. |
|---|---|---|---|---|---|
| *Llama, DeepSeek and Mistral Models* | | | | | |
| Mistral-v0.1-7B | 13.5 | 26.1 | 28.0 | 23.2 | 22.7 |
| ↪ + SimpleRL-Zoo | 21.8 | 28.1 | 34.6 | 30.3 | 28.7 |
| Llama-3.1-8B | 16.1 | 27.1 | 28.7 | 22.7 | 23.6 |
| ↪ + SimpleRL-Zoo | 25.1 | 40.7 | 44.5 | 20.2 | 32.6 |
| DeepSeek-Math-7B | 11.5 | 21.6 | 22.7 | 19.2 | 18.7 |
| ↪ + SimpleRL-Zoo | 16.3 | 47.4 | 45.5 | 27.3 | 34.1 |
| Mistral-Small-24B | 17.4 | 30.9 | 31.7 | 20.2 | 25.0 |
| ↪ + SimpleRL-Zoo | 23.5 | 73.9 | 78.8 | 45.0 | 55.3 |
| *Qwen Series Models* | | | | | |
| Qwen-2.5-0.5B | 9.6 | 23.2 | 24.9 | 24.8 | 20.6 |
| ↪ + SimpleRL-Zoo | 14.4 | 32.1 | 34.6 | 26.3 | 26.8 |
| Qwen-2.5-1.5B | 15.2 | 33.1 | 35.4 | 24.8 | 27.1 |
| ↪ + SimpleRL-Zoo | 20.3 | 42.1 | 45.2 | 28.8 | 34.1 |
| Qwen-2.5-7B | 21.3 | 39.8 | 38.6 | 23.7 | 30.8 |
| ↪ + SimpleRL-Zoo | 25.9 | 49.6 | 47.0 | 29.8 | 38.1 |
| Qwen-2.5-Math-7B | 14.1 | 40.6 | 38.0 | 27.8 | 30.1 |
| ↪ + SimpleRL-Zoo | 17.0 | 55.6 | 56.6 | 35.4 | 41.1 |
| Qwen-2.5-14B | 22.9 | 59.8 | 63.5 | 24.8 | 42.7 |
| ↪ + SimpleRL-Zoo | 29.4 | 76.3 | 79.1 | 50.0 | 58.7 |
| Qwen-2.5-32B | 24.6 | 60.7 | 62.7 | 38.9 | 46.7 |
| ↪ + SimpleRL-Zoo | 31.2 | 79.0 | 82.5 | 49.5 | 60.6 |

Table 2: Detailed performance of various models across IFEVAL, MMLU and GPQA. The blue lines represent the models trained with our recipe.

to 30 absolute points when we vary k up to 128. This suggests that zero RL training is not just reranking the model's output distribution within the top k candidates at a reasonably large range of k (Shao et al., 2024), instead, it enhances the model's fundamental reasoning abilities.

**Growth in Response Length May be Unhealthy:** Response length does not always reflect genuine growth in reasoning. In some cases, unstable training can cause models to generate excessive repetitive content until they hit the context length limit, artificially inflating response length without improving reasoning depth. For example, Figure 4 shows that while most models maintain a low clip ratio – below 5% of the data – when their average stopping length steadily increases, Mistral-7B-v0.1 exhibits a high clip ratio and significant fluctuations in stopping length. Upon closer inspection of its responses, we find that the responses consist of incoherent, mixed-language gibberish,

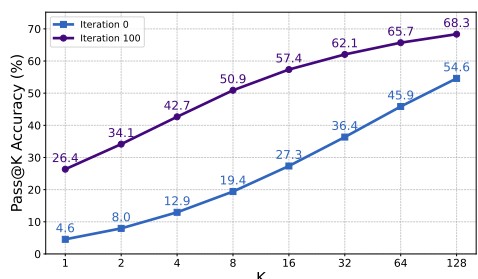

Figure 3: Pass@k of Mistral-24B based on the average results from AIME24 and AMC23.

suggesting that its thinking process is not genuinely expanding. We note that such patterns would not be captured by response length as in Figure 1. These findings indicate that most models demonstrate a meaningful increase in response length. This raises an important question: What exactly do models learn as their thinking time increases? We answer this question next.

## 2.4 The "Aha Moment" – Quantifying Emergence of Reasoning Behaviors

Figure 5 illustrates the reasoning behavior ratio on OlympiadBench during model training. By comparing Figure 5 with Figure 4, we observe that fluctuations in the reasoning behavior ratio effectively account for variations in the average stopped length. Interestingly, we find that different models exhibit entirely distinct trends in reasoning behavior changes.

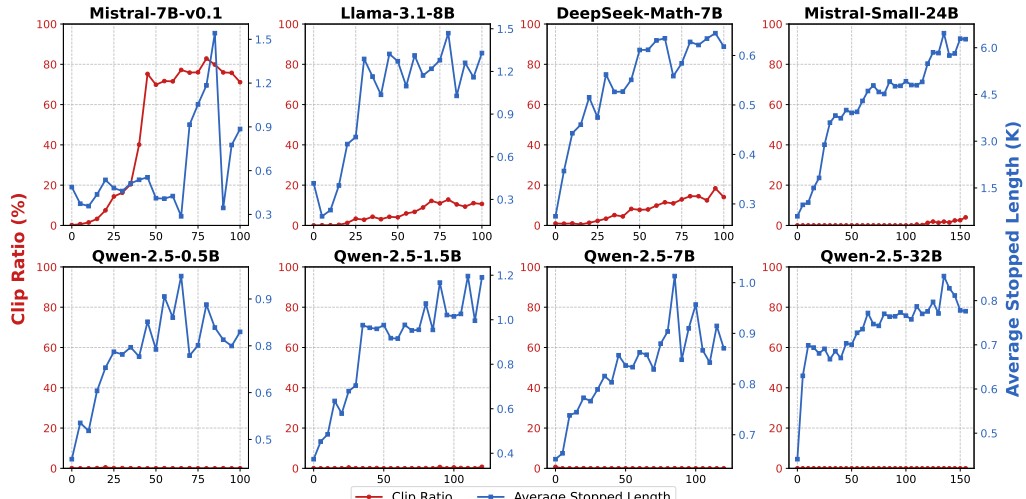

Figure 4: Average clip ratio and stopped length across training iterations for different models. We assess the models every five steps on a variety of math benchmarks, including GSM8K, MATH500, Minerva Math, and OlympiadBench, as well as competition-level benchmarks like AIME24 and AMC23. The red line indicates the clip ratio, while the blue line represents the stopped length. Per-benchmark results are in Figure 12 (Appendix D).

Smaller models, such as Qwen-2.5-0.5B and 1.5B, tend to prioritize learning the "Subgoal Setting" behavior, with its proportion increasing by approximately 4–5 times. Additionally, the proportions of "Verification" and "Enumeration" also show noticeable growth. In contrast, for other base models that inherently possess step-by-step reasoning capabilities, adjustments in "Subgoal Setting" during the RL training process are relatively minor.

DeepSeek-Math-7B, Llama-3.1-8B, and Mistral-Small-24B exhibit substantial increases in the proportions of "Enumeration" and "Verification" behaviors, rising from relatively low initial levels by approximately 3-4 times. This growth correlates closely with their changes in average stopped length, suggesting a shift in reasoning patterns over time. For instance, in Mistral-Small-24B, reflection-oriented behaviors such as "Verification" and "Backtracking" increase dramatically from nearly 0% to approximately 50%, indicating the emergence of reflection behavior from scratch. This shift suggests that the model progressively internalizes verification as part of its reasoning process, offering a promising trajectory for enhancement.

In contrast, Qwen-2.5-7B and 32B demonstrate strong reasoning behaviors from the outset, with minimal changes throughout training. This phenomenon aligns with their slow length adjustments (Figure 1) and suggests that Qwen models inherently possess robust reasoning capabilities. Rather than undergoing a structural shift in their reasoning processes, they primarily benefit from small increases in thinking time, which yield significant performance improvements. Finally, we observe that Mistral-7B-v0.1 consistently exhibits low reasoning behaviors with no noticeable growth, further supporting our earlier analysis in §2.3.

To intuitively illustrate the changes in reasoning behavior, we present examples of Mistral 24B's reasoning before and after training in Figures 22. Comprehensive case studies involving other models are available in Appendix I.3. In Figure 22, we observe that unlike the base model, the zero training model actively attempts to verify if its initial solution is valid by substituting it back into the original expression. Upon recognizing that the first solution does not meet the necessary conditions, the model explicitly initiates a backtracking approach, stating "let's try another possibility," eventually arriving at the correct answer.

## 3 Key Factors Shaping Zero RL Training

In this section, we identify key factors that influence stability and performance during zero RL training, particularly when dealing with early-stage or weaker models. First, we explore how an over-reliance on format rewards restricts exploration. Next, we analyze how

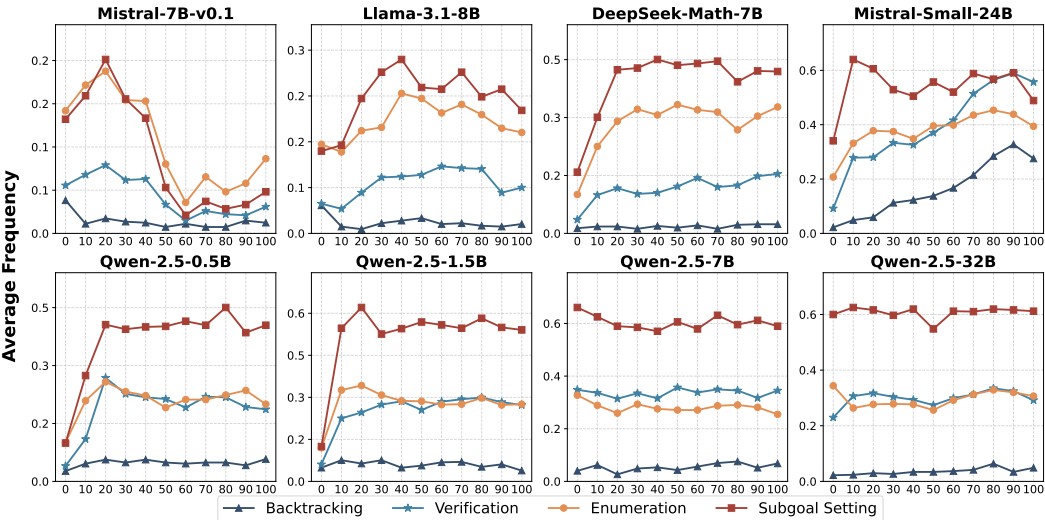

Figure 5: The change in reasoning behavior over the training iterations across all models. As described in §2.2, we use GPT-4o to extract and track shifts in reasoning behaviors on OlympiadBench. We focus on four reasoning-related behaviors: "Backtracking", "Verification", "Subgoal Setting", and "Enumeration".

data difficulty impacts exploratory behavior, illustrating how exposure to varying levels of difficulty shapes the exploration dynamics. We also discuss the impact of exploration-related hyperparameters in Appendix G.

## 3.1 Over-Reliance on Format Rewards

We find that enforcing strict formatting constraints, such as requiring the final answer to be enclosed in a latex command $\backslash boxed\{\}$, can hinder model's freely exploration and ultimately degrades performance. This is because many base models cannot follow the format constraint well in the initial stage, and imposing a format reward will penalize many correct explorations. We compare two reward functions: one without format constraints, which rewards responses solely based on answer correctness (our default design in §2.1), and another that strictly enforces formatting by penalizing responses with a reward of -1 if they fail to adhere to the required format.

Figure 6 illustrates weaker models like Llama-3.1-8B struggle under strict formatting requirements, leading to a rapid increase in response length early in training without performancec improvement. The model expends excessive effort on adhering to the format but fails to learn how to answer correctly, ultimately resulting in model collapse. Figure 6 (Left) further reveals that even stronger models, such as Qwen-2.5-7B, which initially comply with formatting constraints, suffer in later training stages. This includes both performance degradation and a significant reduc-

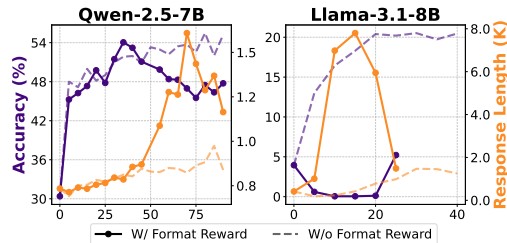

Figure 6: Accuracy and response length with and without format rewards.

tion in CoT length. These findings highlight that: in a zero RL training setting, rather than imposing rigid formatting rules, we should prioritize maintaining response verifiability while allowing sufficient flexibility for exploration.

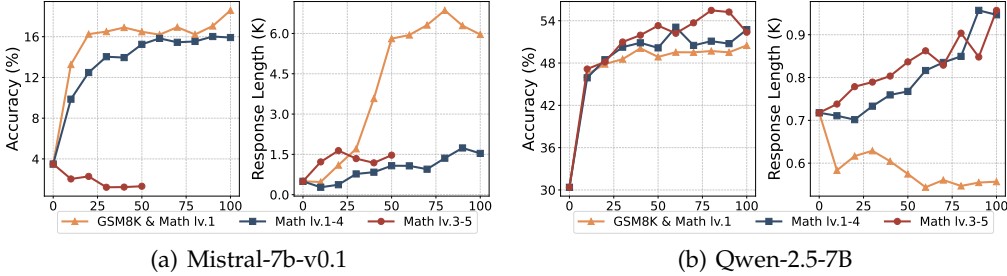

(a) Mistral-7b-v0.1      (b) Qwen-2.5-7B

Figure 7: Comparison of accuracy and response length across different data difficulty levels. We examine three levels of data: Easy (GSM8K and MATH lv.1), Medium (MATH lv.1–4), and Hard (MATH lv.3–5), with each category containing approximately 8,000 problems.

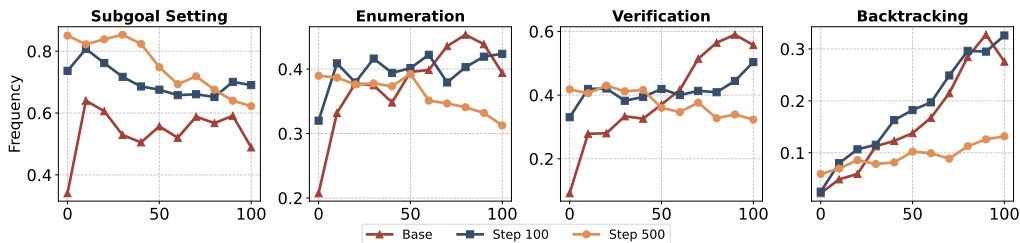

Figure 8: Reasoning behavior ratio over RL training iterations after using different SFT steps as starting points. "Base" refers to the base Mistral-Small-24B model without any SFT, while "Step 100" and "Step 500" represent 100 and 500 steps of SFT, respectively. As described in §2.1, we use GPT-4o to track shifts in reasoning behaviors on OlympiadBench.

## 3.2 Data Difficulty on Exploratory Behavior

Base models exhibit varying performance and CoT behaviors when trained on different RL data. Figure 7 compare the performance of Mistral-7B and Qwen-2.5-7B across Easy, Medium, and Hard datasets. We observe a clear trend: as data difficulty increases, Mistral-7B's performance progressively deteriorates. When faced with high-difficulty data (Hard: MATH levels 3-5), the model struggles to generate responses that receive positive feedback from the reward system. This failure results in a significant increase in response length without any corresponding improvement in accuracy, signaling a breakdown in the training process—often referred to as training collapse. Figure 7 Left demonstrates that Qwen-2.5-7B exhibits a pattern entirely opposite to Mistral-7B-v0.1. Specifically, as dataset difficulty decreases, both the model's average accuracy and response length decline, with the effect being most pronounced on the simplest dataset, where even response length decreases. This finding aligns with our previous analysis of Qwen-2.5-7B in §2.4, reinforcing the notion that Qwen inherently possesses strong reasoning capabilities. To further improve its response length, training should incorporate more challenging datasets to encourage deeper reasoning and extended thinking time. The analysis highlights that zero RL training data must align with the base model's inherent reasoning capabilities.

## 4 Revisiting Traditional SFT for RL-Driven Reasoning Emergence

As base models may not follow instruction well and pose a major challenge for zero RL training, one may wonder a simple SFT stage as a cold start may be helpful to learn to follow instructions well. In this section, we revisit the impact of traditional SFT methods (where the responses are not from long CoT models) as a cold start on RL training performance and reasoning behavior–notably, this was the most commonly used post-training pipeline with RL following an SFT stage, before DeepSeek-R1. Specifically, we use a subset of

the NuminaMath (Li et al., 2024) dataset derived from GSM8K and MATH, [1] containing approximately 15K high-quality short CoT responses. We conduct SFT using Mistral 24B and select models at 100 and 500 training steps as starting points for RL training.

Figure 9 illustrates how model accuracy and output length evolve during RL training when different initial models are used. Our results indicate that starting from SFT models initially boosts performance significantly; however, these models encounter notable limitations in their maximum achievable accuracy and response length compared to starting from the base model during RL training. Crucially, we observe that these limitations become increasingly pronounced as the number of initial SFT steps grows.

To further investigate how initial SFT affects the emergence of reasoning behaviors, we analyze how often specific reasoning behaviors appeared during training at different starting points, as shown in Figure 8. Our analysis reveals that initial SFT negatively impacts the development of critical reasoning behaviors. Specifically, models with 100 SFT steps exhibit reduced upper limits in essential reasoning behaviors such as "enumeration," "verification," and "backtracking," compared to the base model. Even more notably, models with 500 SFT steps experience significant declines in "enumeration" and "verification" behaviors in later training stages, highlighting a detrimental long-term effect of extensive sft on reasoning capabilities.

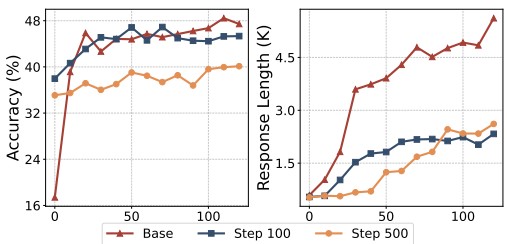

Figure 9: Accuracy and response length averaged on the six benchmarks over RL training iterations after running different SFT steps as starting points.

This prompts a reconsideration of whether traditional SFT inherently restricts model exploration, perhaps highlighting the need for future cold-start strategies to prioritize exploration capacity—whether by incorporating long CoT data (DeepSeek-AI et al., 2025a; Yeo et al., 2025) or designing SFT techniques (Li et al., 2025) that strike a balance between imitation and exploration—to enable sustained improvements in model reasoning performance.

## 5 Conclusion

Our paper demonstrates the effectiveness of zero RL training across a diverse range of base models, yielding significant improvements in accuracy and response length. We provide strong evidence that zero RL training is not merely reranking, but rather a genuine enhancement. Furthermore, we identify key factors such as reward design, data difficulty, and models' inherent abilities that shape the emergence of advanced reasoning behaviors. Our findings also indicate that starting RL training from models with traditional SFT may limit the development of advanced reasoning behaviors. Overall, our work highlights key factors for effective zero RL training and offers insights for future model improvements.

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

# A Detailed Background: "Zero RL Training"

In our study, we follow the zero RL training recipe in DeepSeek-AI et al. (2025a) using various open base models, employing the GRPO algorithm (Shao et al., 2024). Here, zero RL training refers to reinforcement learning directly from the base model without any prior supervised fine-tuning (SFT). GRPO optimizes computational efficiency by eliminating the need for a separate value model; instead, it directly utilizes group-normalized rewards to estimate advantages. For a query $q$ and a set of responses $O = \{o_1, o_2, \ldots, o_G\}$ sampled from the old policy model $\pi_{\text{old}}$, we adopt a token-level, length-rectified GRPO objective to optimize the policy model $\pi$:[2]

$$\mathcal{J}_{\text{GRPO}}(\theta) = \underbrace{\frac{1}{\sum_{i=1}^{G} |o_i|} \sum_{i=1}^{G} \sum_{t=1}^{|o_i|} \min\left[r_{i,t}(\theta)\hat{A}_i, \text{clip}\left(r_{i,t}(\theta); 1-\epsilon, 1+\epsilon\right)\hat{A}_i\right]}_{\text{Clipped policy update}} - \underbrace{\beta \, \mathbb{D}_{\text{KL}}[\pi_\theta \parallel \pi_{\text{ref}}]}_{\text{KL penalty}}$$

$$\text{where } r_{i,t}(\theta) = \frac{\pi_\theta(o_{i,t} \mid q, o_{i,<t})}{\pi_{\theta_{\text{old}}}(o_{i,t} \mid q, o_{i,<t})} \tag{1}$$

where $\pi_{\text{ref}}$ represents the reference model, and the term $\mathbb{D}_{KL}$ introduces a KL divergence constraint to limit how much the model can deviate from this reference. The advantage estimate $\hat{A}_i$ measures how much better the response $o_i$ is compared to the average response, which is computed using a group of rewards $\{r_1, r_2, \ldots, r_G\}$ for the responses in set $O$:

$$\hat{A}_i = \frac{r_i - \text{mean}(\{r_1, r_2, \ldots, r_G\})}{\text{std}(\{r_1, r_2, \ldots, r_G\})} \tag{2}$$

# B Detailed Experimental Setup

## B.1 Dataset

To keep the training recipe simple, we select training data exclusively from the GSM8K (Cobbe et al., 2021) and MATH (Hendrycks et al., 2021) datasets. For the MATH dataset, following prior studies (Lightman et al., 2023; Wang et al., 2023; Sun et al., 2024), we reserve the MATH500 subset as the test set, uniformly sample an additional 500 problems for validation, and combine the remaining 4,000 test problems with the original 7,500 training problems to form our training set. Each example in the MATH dataset is originally labeled with a difficulty level ranging from 1 to 5. In our experiments, we find that data difficulty is critical for successful zero RL (§3.2) and it is necessary to use data that aligns with the model's capability. To investigate this phenomenon, we categorize the data into three difficulty levels: Easy (GSM8K and MATH lv.1), Medium (MATH lv.1–4), and Hard (MATH lv.3–5), with each category containing roughly 8,000 problems. For our main training runs, we use Easy for LLama-3.1-8B, Mistral- v0.1-7B, and DeepSeek-Math-7B; Medium for Qwen-2.5-0.5B; Hard for Mistral-Small-24B, Qwen-2.5-Math-7B, and Qwen-2.5-1.5B/7B/14B/32B, and we will report ablation study on data difficulty in §3.2.

## B.2 Reward

We use a rule-based reward function that assigns rewards solely based on the correctness of the generated response: a correct final answer receives a reward of +1, while an incorrect one receives a reward of 0. Recent studies (Luo et al., 2025; Chen et al., 2025) often incorporate format-based rules into reward calculation, encouraging the model to follow specific output

---

[2]The original GRPO objective has a length normalization term that introduces length biases. We remove the length normalization term similar to concurrent works (Yu et al., 2025; Liu et al., 2025) – this length-rectified objective was the default implementation of GRPO in our adapted codebase, verl (Sheng et al., 2024).

formats. However, we find that this approach may hinder the model's exploration and ultimately harm its performance particularly for the base models which struggle with following the format in the initial stage, as detailed in §3.1.

### B.3 Models

We conduct zero RL training experiments on Llama-3.1-8B (Dubey et al., 2024), DeepSeek-Math-7B (Shao et al., 2024), Mistral-v0.1-7B (Jiang et al., 2023), Mistral-Small-24b-Base-2501 (Mistral AI, 2025), and Qwen-2.5 (0.5B, 1.5B, 7B, 14B, 32B) (Yang et al., 2024a). As we perform experiments for a variety of models, under extremely simple settings with small, simple datasets and only correctness reward, we refer to our obtained models as *SimpleRL-Zoo* to represent a simple training recipe for a zoo of open base models. For models with weaker instruction-following capabilities (Llama-3.1-8B, Mistral-v0.1-7B, and Qwen-2.5-0.5B/1.5B), we employ simpler prompts (Chern et al., 2023) requiring only step-by-step reasoning. For models with stronger instruction-following abilities, we use more complex prompts (Yang et al., 2024a) that require the final answers to be placed in boxes. In our preliminary experiments, we observe that using complex prompts with models that have weak instruction-following capabilities often results in large amounts of irrelevant or nonsensical content being generated early in training, leading to instability. The content of simpler prompts and more complex prompts is shown in Figure 10 in Appendix.

### B.4 Benchmark

We evaluate performance on standard mathematical reasoning benchmarks, including GSM8K (Cobbe et al., 2021), MATH 500 (Hendrycks et al., 2021), Minerva Math (Lewkowycz et al., 2022), and OlympiadBench (He et al., 2024), as well as on competition-level benchmarks such as AIME 2024 and AMC 2023.

### B.5 Training and Evaluation Details

We train our models using the verl (Sheng et al., 2024) framework. And we typically use the same set of hyperparameters to train and evaluate all models in the SimpleRL-Zoo series in default main experiment setting. We use a prompt batch size of 1,024 and generate 8 rollouts per prompt, with a maximum rollout length of 8,192 tokens. Training is performed using a mini-batch size of 256. The default sampling temperature is set to 1.0, and the clip ratio is 0.2. For models ranging from 0.5B to 14B parameters, we use a KL loss coefficient of 1e-4. For models larger than 14B, the KL loss coefficient is set to 1e-3. We build our evaluation script based on Yang et al. (2024b), using a temperature of 1.0 and a maximum generation length of 16K tokens. To ensure consistency, we adopt the same prompt template used during training. For most benchmarks, we report pass@1 results. However, for AIME 2024, which contains fewer problems, we report both pass@1 and average accuracy (avg@32), computed over 32 generated samples per problem.

## C Detailed Evaluation Metrics

**Reasoning Behavior Ratio:** To better understand the model's reasoning patterns throughout the training process, we adopt the cognitive behavior framework proposed by Gandhi et al. (2025) and use GPT-4o (Hurst et al., 2024) to identify reasoning-related behaviors, including "Backtracking", "Verification", "Subgoal Setting", and "Enumeration". We report the ratio of responses that contain such cognitive behaviors. While some recent studies suggest tracking reflection behavior using related keywords (Yeo et al., 2025; Xie et al., 2025) as monitoring signals, we argue that these keywords only exhibit only a weak correlation with high-level reasoning patterns like reflection and verification. As a result, they fail to adequately capture the development of these reasoning processes. Further details can be found in Appendix I.

**Clip Ratio:** In the early stages of training, the base model exhibits weak instruction-following ability and often fails to stop appropriately, resulting in irrelevant or excessively

> **Simple Prompt**
>
> Question:
> {input}
> Answer:
> Let's think step by step.

> **Complex Prompt**
>
> <|im_start|>system
> You are a helpful assistant.<|im_end|>
> <|im_start|>user
> {input}
> Please reason step by step, and put your final answer
> within \\boxed{}.<|im_end|>
> <|im_start|>assistant
> {output}

Figure 10: Comparison between simple prompts and more complex prompts.

long outputs. After training collapses, the model may also generate repetitive or overly extended responses. Since the model has a fixed maximum context length, such outputs may be truncated during both training and evaluation. To monitor this issue, we define the proportion of truncated outputs as the "Clip Ratio".

**Average Stopped Length:** Generations that are truncated often result from issues such as repetitive patterns or incomplete reasoning, which typically do not contribute to effective trajectories. To account for this factor, we introduce a new metric to track the average length of responses that are stopped under normal conditions. It is a more reliable metric to consider only valid responses, thereby eliminating the interference caused by unstopped responses.

**Pass@k Accuracy:** We track the pass@k accuracy, which represents the percentage of questions for which at least one correct response is obtained when sampling k responses per question. Pass@k serves as an indicator of the model's exploration capabilities and is particularly relevant for RL, as it reflects the model's ability to generate responses that can achieve a positive reward. Previously, some researchers believed that RL training might merely reorder responses within the original model distribution, as evidenced by the lack of improvement in pass@k accuracy following RL training (Shao et al., 2024).

## D  Detailed Result of SimpleRL

Following the setup described in Section 2.1, we perform "zero training" on various base models. The trained models are then evaluated on multiple benchmarks, including GSM8K, MATH 500, Minerva Math, OlympiadBench, AIME2024, and AMC2023. The average results across all these benchmarks are presented in Figures 1 and 4. In this section, we provide more detailed results. Figure 11 illustrates the trends in accuracy and response length, while Figure 12 shows the trends in clip ratio and stopped length.

## E  Quantitative Behavior Validation

We assess the consistency between GPT-4o labeled reasoning behaviors and human annotations by having human experts annotate 105 model outputs. Table 3 below presents the prediction rates and agreement rate. The prediction rate reflects how frequently each reasoning behavior is identified, while the agreement rate is the proportion of data on which the labelers (Human and GPT-4o) make the same prediction.

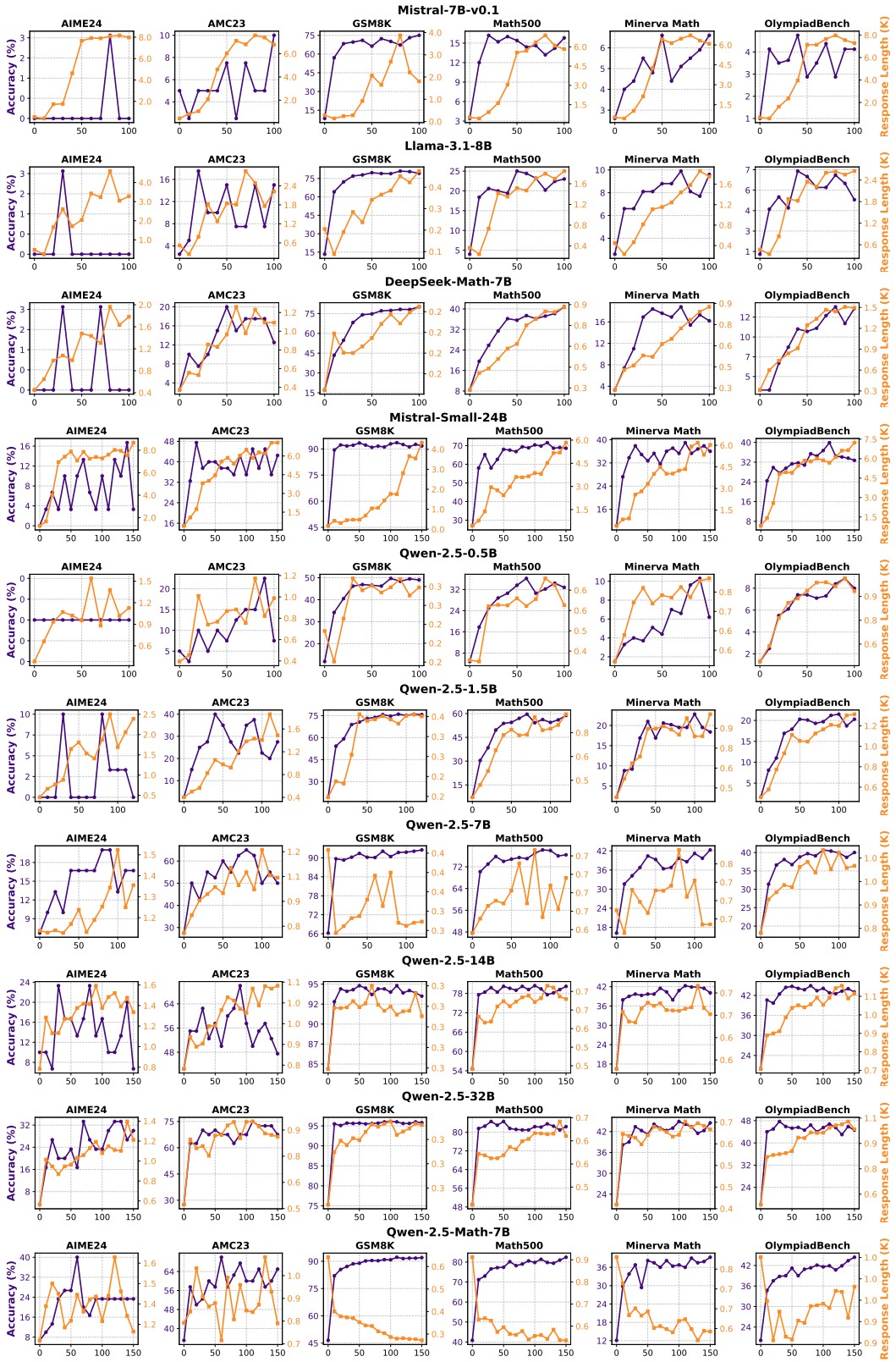

Figure 11: A detailed evaluation of accuracy and response length throughout the training steps for various models. The x-axis represents the training steps, with the purple line showing the accuracy trend and the yellow line depicting the response length.

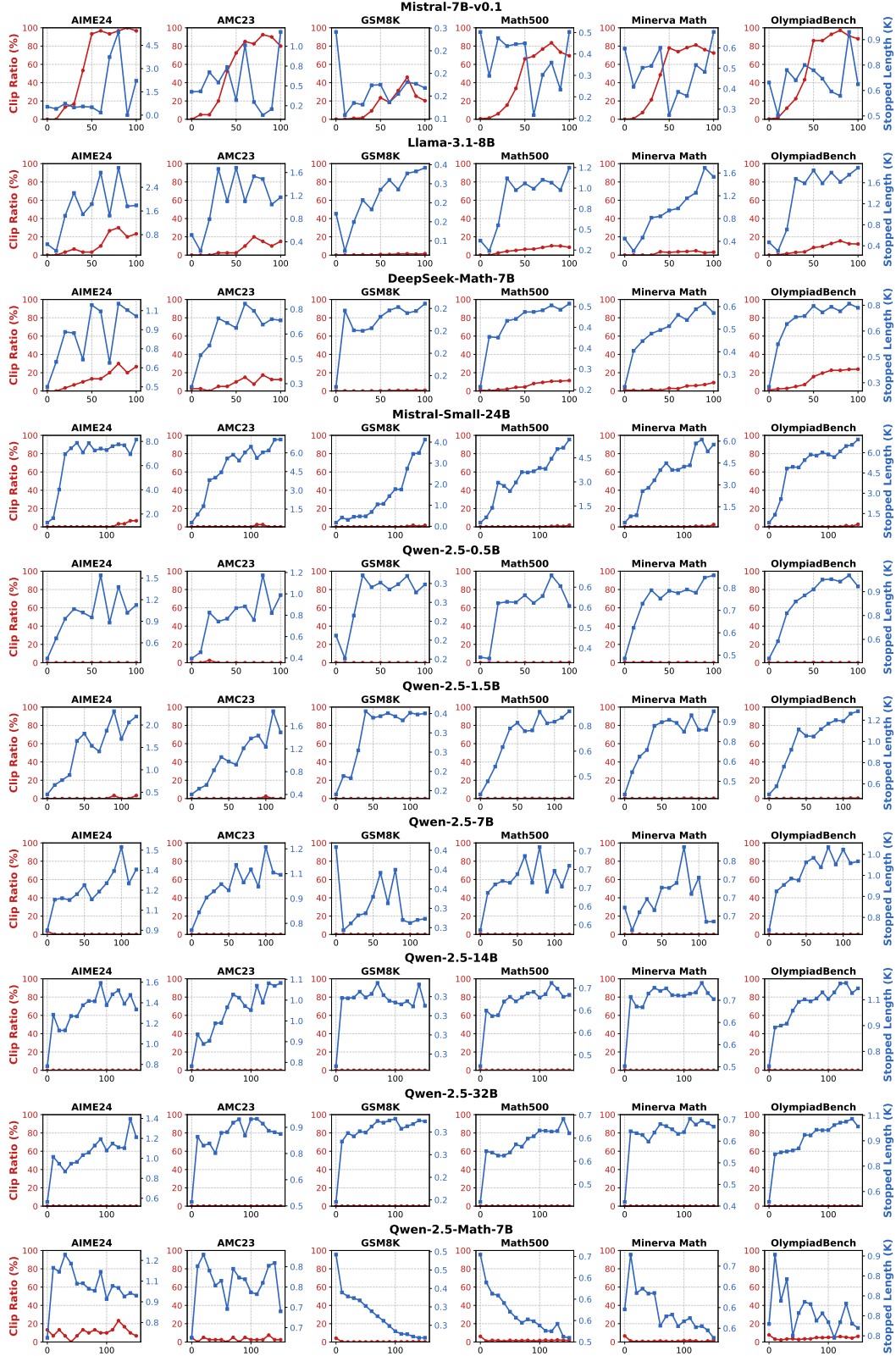

Figure 12: A detailed evaluation of clip ratio and stopped length throughout the training steps for various models. The x-axis represents the training steps, with the red line showing the clip ratio trend and the blue line depicting the average stopped length.

| Behavior | Score by GPT-4o (%) | Score by Human (%) | Raw Agreement (%) |
|---|---|---|---|
| Verification | 78.10% (82/105) | 85.71% (90/105) | 90.48% (95/105) |
| Backtracking | 33.33% (35/105) | 35.24% (37/105) | 98.10% (103/105) |
| Subgoal Setting | 66.67% (70/105) | 74.29% (78/105) | 90.48% (95/105) |
| Enumeration | 61.90% (65/105) | 63.81% (67/105) | 94.29% (99/105) |

Table 3: The consistency between GPT-4o labeled reasoning behaviors and human annotations

| Init Model | GSM8K | MATH 500 | Minerva Math | Olympiad Bench | AIME24 (pass@1) | AMC23 | Avg. |
|---|---|---|---|---|---|---|---|
| 0 Step | 92.0 | 70.6 | 36.8 | 36.6 | 16.7 | 45.0 | 49.6 |
| 10 Step | 93.0 | 69.4 | 39.7 | 32.3 | 10.4 | 44.1 | 48.2 |
| 20 Step | 92.6 | 65.2 | 34.2 | 30.7 | 6.7 | 38.4 | 44.6 |
| 200 Step | 90.3 | 59.0 | 31.6 | 23.3 | 2.1 | 26.9 | 38.9 |
| 1000 Step | 88.9 | 48.8 | 27.6 | 20.7 | 2.5 | 18.1 | 34.4 |
| 2000 Step | 89.8 | 49.0 | 23.2 | 18.1 | 0.8 | 20.3 | 33.5 |
| 4000 Step | 87.7 | 52.0 | 23.5 | 17.2 | 2.1 | 21.6 | 34.0 |

Table 4: Experimental results from multiple Mistral-Small-24B models, each fine-tuned with a different number of SFT steps on a general SFT dataset for RL. The "number of steps" refers to the number of SFT steps applied. The reported benchmarks reflect the performance metrics on various evaluation benchmarks, measured using the model that achieved the best average performance after 100 iterations of reinforcement learning training.

Our results indicate a generally good level of agreement between GPT-4o and human annotations. However, GPT-4o tends to be more conservative when labeling certain behaviors such as Verification and Subgoal Setting. Upon closer examination, we observe that in cases with long CoT containing multiple reasoning behaviors, the model often favors labeling more obvious behaviors like Enumeration, while overlooking subtler ones.

## F   Impact of General SFT on the Performance of Reinforcement Learning

We also investigated the general SFT setting beyond math-related datasets. In this setup, we first conducted SFT on Mistral-Small-24B using the widely adopted OpenHermes-2.5 dataset.[3] We implement with LLaMA-Factory (Zheng et al., 2024) and adopt common hyperparameters of SFT, including 512 examples per batch with a constant learning rate of 1e-5. For consistency with our other experiments, we fine-tuned the model using the Qwen chat template. After SFT, we preserved multiple checkpoints at different training steps, and nearly 800 steps correspond to 1 epochs on the SFT dataset. We then performed reinforcement learning on these models using identical hyperparameters as in our zero-RL training experiments.

Table 4 presents our findings, with performance reported as the best results achieved during RL training up to 100 iterations. The results demonstrate an inverse relationship between SFT steps and subsequent RL performance: models with more SFT steps showed diminished performance after RL training. While the average performance after 10 SFT steps remained comparable to the base model, it still exhibited some negative effects. More significantly, models with more than 20 steps showed substantially reduced RL potential. Therefore, we conclude that RL training produces the best performance gain when applied directly to the base model without any supervised fine-tuning, i.e., the zero RL training.

---

[3]https://huggingface.co/datasets/teknium/OpenHermes-2.5

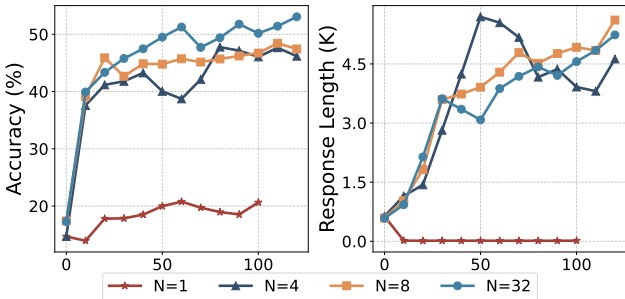

Figure 13: Comparison of accuracy and response length using different sampling numbers N = 1, 4, 8, 32. The training data is the Hard part (MATH lv.3–5) with the same setting in main results, as described in § 2.1.

## G  Impact of Exploration-Related Hyperparameters

In this section, we examine the effects of exploration-related hyperparameters on "zero-training." Drawing inspiration from Zeng et al. (2025b); Liu et al. (2024), we focus on two key factors: sampling size (the number of responses per query) and sampling temperature.

**Sampling Size:**  We examine how varying sampling sizes $N \in \{1, 4, 8, 16, 32\}$ influence the training process using the Mistral 24B model; these results are presented in Figure 13. Our analysis reveals a clear trend: as $N$ increases, the model's average performance notably improves, and variability in response lengths becomes significantly more stable. For example, after 100 training steps, the scenario with $N = 32$ achieves an average accuracy approximately 6 points higher than that with $N = 8$. Conversely, smaller sampling sizes ($N = 1$ and $N = 4$) cause training instability and potential collapse, indicated by rapid growth in generated length without corresponding accuracy improvements. We hypothesize that larger sample sizes enable the model to explore a broader and more diverse training space, which stabilizes advantage estimation and sustains continuous performance improvement.

**Sampling Temperature:**  We conduct research on Qwen-2.5-0.5B to analyze the impact of sampling temperature during both training and evaluation on model performance. The results, presented in Figure 16 , indicate that training with higher temperatures generally leads to better average performance. For instance, models trained with temperatures of 1.0 and 1.2 outperform those trained with 0.8 and 0.6. Additionally, we find that the optimal evaluation temperature depends on the training temperature. Specifically, models trained at higher temperatures require higher sampling temperatures during evaluation, as using greedy sampling often results in repetitive outputs. Conversely, models trained at lower temperatures perform best when evaluated with lower sampling temperatures.

## H  SimpleRL-Zoo For Qwen2.5-Math-7B

In this section, we conduct experiments on Qwen2.5-Math-7B (Yang et al., 2024a) using the "hard part" data, as described in § 2.1, which consists of only 8K examples from MATH lv3-5. We apply both the PPO and GRPO algorithms to train our base model, and the overall evaluation results across training steps are shown in Figure 14. The final performance and response length for both algorithms converge to similar values, with GRPO slightly outperforming PPO. While the performance continues to improve, the response length does not exhibit a similar trend. Specifically, the stopping length for both algorithms remains relatively unchanged, and fluctuations in the average response length are primarily attributed to changes in the clip ratio. There are two main reasons for this behavior: First, the maximum context length for Qwen2.5-Math-7B is 4K, which is limited compared to other models with context lengths exceeding 8K, leading to a high clip ratio. Second, as a math-specific model, Qwen2.5-Math-7B already performs very well on MATH, the dataset

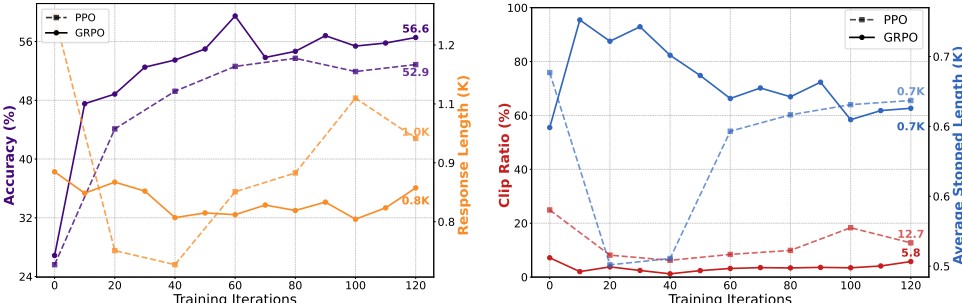

Figure 14: Comparison of accuracy and response length between PPO and GRPO on Qwen2.5-Math-7B. The base model is trained using 8K examples from MATH lv3-5, with the same settings described in § 2.1.

we used for training, so it may not face enough challenge to further extend its response length. Therefore, we hypothesize that more challenging data might be needed to push this capable model further.

# I   Reasoning Behavior Analysis

We apply  Gandhi et al. (2025)'s cognitive behavior framework to perform a detailed analysis of how model reasoning behaviors change during "zero training." We first describe our analysis setup, then compare reflection keyword tracking against this framework to monitor reflective behaviors. Finally, we use case studies to illustrate how the reasoning behaviors of various models evolve during training.

## I.1   Setup

We use GPT4-o to identify and analyze the following key reasoning behaviors exhibited in the model's responses, with the prompt shown in Figure 17:

(1) **Backtracking**: The model actively identifies errors during response generation and explicitly revises previously used methods.

(2) **Verification**: The model systematically checks intermediate results to ensure correctness.

(3) **Subgoal Setting**: The model decomposes complex problems into smaller, manageable steps.

(4) **Enumeration**: The model exhaustively considers multiple cases or possibilities to solve problems.

Note that we replaced "Backward Chaining" with "Enumeration," as the former was not relevant to our task.

## I.2   Comparison of Different Reasoning Behavior Tracking Methods

Using DeepSeek Math's "zero-training" process as an example, we compare two different methods for monitoring reasoning behavior. The first method tracks the occurrence of specific keywords in the model's responses, such as "recheck," "rethink," "try again," "wait," "alternatively," "retry," and "however." The second method employs  (Gandhi et al., 2025)'s cognitive framework for evaluation. Figure 15 illustrates the observed changes in reasoning behavior according to these two approaches. During the training process, we observe that the proportion of specified keywords in the DeepSeek math model's responses remains consistently low, exhibiting minimal variation. Conversely, reasoning behaviors identified by the cognitive framework demonstrate a significant upward trend.

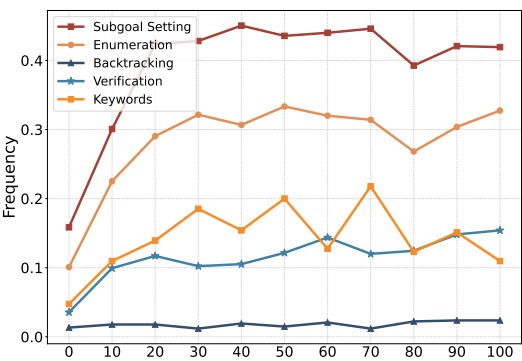

Figure 15: Changes in reflection behavior iden-
tified by different methods.

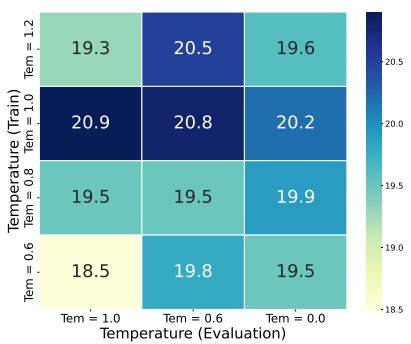

Figure 16: Impact of training and
evaluation temperatures on Qwen-
2.5-0.5b's average final performance
(x-axis: evaluation temp, y-axis:
training temp).

To understand this intriguing discrepancy, we manually review the reasoning behaviors
recorded by the cognitive framework. Our analysis reveals that many of these reasoning
behaviors do not necessarily involve the predefined keywords. For instance, in Figure 18,
the observed reasoning behaviors include Verification and Backtracking, neither of which
contains the specified keywords. This indicates that keywords alone cannot effectively
distinguish or capture the nuanced differences between such behaviors. Similarly, in
Figure 19, the reasoning process involves implicit verification steps, including recalculating
intermediate results such as the dot product and magnitudes before determining the cosine
of the angle. Again, these subtle verification steps are not represented by the designated
keywords. In Figure 21, the reasoning involves considering multiple possible scenarios or
outcomes. This type of exploratory reasoning is also inadequately captured by keyword-
based approaches. These examples collectively illustrate that relying solely on keyword
presence is insufficient for accurately identifying and differentiating complex reasoning
behaviors within model responses.

### I.3   Reasoning Behavior Variations Across Different Models

We present cases illustrating notable improvements in model reasoning behavior during
training (Figure 5). Specifically, these improvements are demonstrated in the following
models: Mistral 24B (Figure 22 and Figure 23), Qwen 2.5-0.5B (Figure 24, Figure 25 and
Figure 26), Qwen 2.5-1.5B (Figure 27 and Figure 28), DeepSeek-math-7B-base (Figure 18,
Figure 19, Figure 20 and Figure 21), and Llama 3.1-8B (Figure 29 and Figure 30).

## Prompt for Identifying and Analyzing Reasoning Behaviors

Below is a chain-of-reasoning generated by a Language Model when attempting to solve a math problem. Evaluate this chain-of-reasoning to determine whether it demonstrates beneficial problem-solving behaviors that deviate from typical linear, monotonic reasoning patterns commonly observed in language models.

<start_of_reasoning>
{input}
<end_of_reasoning>

Specifically, actively identify and emphasize beneficial behaviors such as:
(1) Backtracking: Explicitly revising approaches upon identifying errors or dead ends
 (e.g., "This approach won't work because...").

(2) Verification: Systematically checking intermediate results or reasoning steps
(e.g., "Let's verify this result by...").

(3) Subgoal Setting: Breaking down complex problems into smaller, manageable steps
(e.g., "To solve this, we first need to...").

(4) Enumeration: Solving problems by exhaustively considering multiple cases or possibilities.

Additionally, remain attentive to and encourage the identification of other beneficial behaviors not explicitly listed here, such as creative analogies, abstraction to simpler cases, or insightful generalizations.

Important:
Clearly specify each beneficial behavior you identify.
Provide explicit examples from the reasoning chain.
If no beneficial behaviors are observed, explicitly return an empty list.
Provide your evaluation clearly, formatted as follows:

```json
{
  "behaviour": "",
  "example": ""
}
```

Figure 17: Prompt for identifying and analyzing reasoning behaviors.

**Question**

The average of three consecutive multiples of 3 is $a$.\n\nThe average of four consecutive multiples of 4 is $a+27$.\n\nThe average of the smallest and largest of these seven integers is 42 .\n\nDetermine the value of $a$.

**Base Solution**

find all multiples of 3 and 4, put them into a bracket with the same name,\n\n<|im_start|>system\nThe first seven multiples are [3,6,9,12,15,18,21]

**Zero Solution**

Let's reason step by step:\n\nThe average of three consecutive multiples of 3 is a.\nLet's call the three consecutive multiples of 3 as 3x, 3x+3, and 3x+6, where x is an integer.\nThe average of 3x, 3x+3, and 3x+6 is a.\n(3x + 3x+3 + 3x+6) / 3 = a\n9x + 9 = 3a\n3x + 3 = a\nSo, the three consecutive multiples of 3 are 3x, 3x+3, and 3x+6, where 3x+3 = a.\n\nThe average of four consecutive multiples of 4 is a+27.\nLet's call the four consecutive multiples of 4 as 4y, 4y+4, 4y+8, and 4y+12, where y is an integer.\nThe average of 4y, 4y+4, 4y+8, and 4y+12 is a+27.\n(4y + 4y+4 + 4y+8 + 4y+12) / 4 = a+27\n16y + 24 = 4(a+27)\n4y + 6 = a+27\n4y = a+21\ny = (a+21)/4\nSo, the four consecutive multiples of 4 are 4y, 4y+4, 4y+8, and 4y+12, where y = (a+21)/4.\n\nThe average of the smallest and largest of these seven integers is 42.\nThe smallest integer is 3x, and the largest integer is 4y+12.\nThe average of 3x and 4y+12 is 42.\n(3x + 4y+12) / 2 = 42\n3x + 4y+12 = 84\n3x + 4((a+21)/4) + 12 = 84\n3x + a+21 + 12 = 84\n3x + a+33 = 84\n3x + a = 51\n3x + 3x+3 = 51\n6x + 3 = 51\n6x = 48\nx = 8\nSo, 3x = 24, 3x+3 = 27, and 3x+6 = 30.\n4y = (a+21)/4 = (27+21)/4 = 48/4 = 12\n4y+4 = 12+4 = 16\n4y+8 = 12+8 = 20\n4y+12 = 12+12 = 24\nSo, the seven integers are 24, 27, 30, 16, 20, 24, and the average of the smallest and largest is (24+24)/2 = 48/2 = 24.\n\nThe average of three consecutive multiples of 3 is a, which is 27.\nThe average of four consecutive multiples of 4 is a+27, which is 27+27 = 54.\n\nSo, the three consecutive multiples of 3 are 24, 27, and 30, and the four consecutive multiples of 4 are 16, 20, 24, and 28.\n\nThe smallest integer is 16, and the largest integer is 30.\nThe average of the smallest and largest is (16+30)/2 = 46/2 = 23.

Let's check if the average of the smallest and largest is 42:
The smallest integer is 16, and the largest integer is 30.
The average of 16 and 30 is (16+30)/2 = 46/2 = 23.

So, the average of the smallest and largest is not 42.
Let's try again:
The smallest integer is 24, and the largest integer is 30.

The average of 24 and 30 is (24+30)/2 = 54/2 = 27.\nSo, the average of the smallest and largest is 27, which is a.\n\nSo, the value of a is 27.\n\n\boxed{27}\n

Figure 18: A comparison of DeepSeek-Math-7B's "Backtracking" behavior before and after zero RL training. Here, "base solution" represents the response of the DeepSeek-Math-7B base model, while "zero solution" represents the response of the model after training.

**Question**

The set of points $(x,y,z)$ that satisfy\n\\[2x = 3y = -z\\]is a line.\n\nThe set of points $(x,y,z)$ that satisfy\n\\[6x = -y = -4z\\]is another line.\n\nFind the angle between these lines, in degrees.

**Base Solution**

I'm not sure what you mean by \"the set of points.\". A set of points is not a line.\ When you say to \"find the angle,\" for what purpose? Do you need to understand the relationship between the two lines, or is it just a math problem?

**Zero Solution**

Let's reason step by step:\n\nThe set of points (x,y,z) that satisfy 2x = 3y = -z is a line.\n\nLet's find the direction vector of this line.\n\nFrom 2x = 3y, we get x = 3y/2.\n\nFrom 2x = -z, we get z = -2x = -3y.\n\nLet's let y = 2, then x = 3 and z = -6.\n\nSo a point on this line is (3,2,-6).\n\nThe direction vector of this line is (3-0,2-0,-6-0) = (3,2,-6).\n\nThe set of points (x,y,z) that satisfy 6x = -y = -4z is another line.\n\nLet's find the direction vector of this line.\n\nFrom 6x = -y, we get y = -6x.\n\nFrom 6x = -4z, we get z = -6x/4 = -3x/2.\n\nLet's let x = 2, then y = -12 and z = -3.\n\nSo a point on this line is (2,-12,-3).\n\nThe direction vector of this line is (2-0,-12-0,-3-0) = (2,-12,-3).\n\nLet's find the angle between these lines.

The dot product of the direction vectors is (3,2,-6) · (2,-12,-3) = 6 - 24 + 18 = 0.

The magnitude of the direction vector of the first line is sqrt(3^2 + 2^2 + (-6)^2) = sqrt(9 + 4 + 36) = sqrt(49) = 7.\n\nThe magnitude of the direction vector of the second line is sqrt(2^2 + (-12)^2 + (-3)^2) = sqrt(4 + 144 + 9) = sqrt(157).
The cosine of the angle between the lines is the dot product of the direction vectors divided by the product of the magnitudes of the direction vectors.

The cosine of the angle between the lines is 0/(7*sqrt(157)) = 0.

The angle between the lines is arccos(0) = 90 degrees.\n\nSo the angle between these lines is 90 degrees.\n\nLet's put the final answer within \\boxed{90}.\n\nSo the final answer is \\boxed{90}.\n

Figure 19: A comparison of DeepSeek-Math-7B's "Verification" behavior before and after zero RL training. Here, "base solution" represents the response of the DeepSeek-Math-7B base model, while "zero solution" represents the response of the model after training. Here involves implicit verification steps, including recalculating intermediate results such as the dot product and magnitudes before determining the cosine of the angle.

**Question**

For a positive integer $a$, define a sequence of integers $x_{1}, x_{2}, \ldots$ by letting $x_{1}=a$ and $x_{n+1}=2 x_{n}+1$ for $n \geq 1$. Let $y_{n}=2^{x_{n}}-1$. Determine the largest possible $k$ such that, for some positive integer $a$, the numbers $y_{1}, \ldots, y_{k}$ are all prime.

**Base Solution**

If $x_{n+1}=2 x_{n}+1$, then $y_{n+1}=2 x_{n+1}-1=(2^{x_{n+1}-x_{n}}-1)(2^{x_{n}}+1)$. So, $y_{n+1}<y_{n}$ if and only if $x_{n+1}-x_{n}>1$, which is true if and only if $x_{n}=2^{i}-1$ for some $i$. Similarly, $y_{n+1}<y_{n}-2$ if and only if $x_{n+1}-x_{n}>2$, which is true if and only if $x_{n}=2^{i}-2$ for some $i$.

**Zero Solution**

Let's reason step by step.\nLet $a$ be a positive integer.\nDefine a sequence of integers $x_{1}, x_{2}, \ldots$ by letting $x_{1}=a$ and $x_{n+1}=2 x_{n}+1$ for $n \geq 1$.\nLet $y_{n}=2^{x_{n}}-1$.\nWe have $x_{1}=a$ and $x_{n+1}=2 x_{n}+1$ for $n \geq 1$.\nThen $x_{2}=2 x_{1}+1=2a+1$, $x_{3}=2 x_{2}+1=2(2a+1)+1=4a+3$, and so on.\nThus, the sequence $x_{n}$ is $a, 2a+1, 4a+3, 8a+7, \ldots$.\nLet's calculate $y_{n}=2^{x_{n}}-1$.\nWe have $y_{1}=2^{x_{1}}-1=2^{a}-1$, $y_{2}=2^{x_{2}}-1=2^{2a+1}-1$, $y_{3}=2^{x_{3}}-1=2^{4a+3}-1$, and so on.\nLet's find the largest possible $k$ such that, for some positive integer $a$, the numbers $y_{1}, \ldots, y_{k}$ are all prime.\nWe have $y_{1}=2^{a}-1$.\nIf $a=1$, then $y_{1}=2^{1}-1=1$, which is not prime.\nIf $a=2$, then $y_{1}=2^{2}-1=3$, which is prime.\nIf $a=3$, then $y_{1}=2^{3}-1=7$, which is prime.\nIf $a=4$, then $y_{1}=2^{4}-1=15$, which is not prime.\nThus, the smallest possible value of $a$ for which $y_{1}$ is prime is $a=2$.

Let's check $y_{2}=2^{x_{2}}-1=2^{2a+1}-1=2^{2*2+1}-1=2^{5}-1=31$, which is prime. Let's check $y_{3}=2^{x_{3}}-1=2^{4a+3}-1=2^{4*2+3}-1=2^{11}-1=2047=23*89$, which is not prime.

Thus, the largest possible $k$ such that, for some positive integer $a$, the numbers $y_{1}, \ldots, y_{k}$ are all prime is $k=2$.
The final answer is \boxed{2}.

Figure 20: A comparison of DeepSeek-Math-7B's "Verification" behavior before and after zero RL training. Here, "base solution" represents the response of the DeepSeek-Math-7B base model, while "zero solution" represents the response of the model after training. This demonstrates more explicit verification, including key phrases like "Let's check".

**Question**

The expression $10x^2-x-24$ can be written as $(Ax-8)(Bx+3),$ where $A$ and $B$ are integers. What is $AB + B$?

**Base Solution**

Let's start by expanding $(Ax-8)(Bx+3)$:
\\begin{align*}\n(Ax-8)(Bx+3) &= ABx^2 + 3Ax - 8Bx - 24\\\\\\n&= ABx^2 + (3A-8B)x - 24\n\\end{align*}\nWe can compare this expression to the original expression $10x^2-x-24$ and see that $A = 10$ and $3A - 8B = -1$. Therefore, B must be 4 since 30-8B is -1.\nSo, the answer is $10\\times4+4 = \\boxed{44}$.

**Zero Solution**

Let's reason step by step:
The expression $10x^2-x-24$ can be written as $(Ax-8)(Bx+3)$, where A and B are integers.
Expanding $(Ax-8)(Bx+3)$, we get:\n$(Ax-8)(Bx+3) = ABx^2 + 3Ax - 8Bx - 24
= ABx^2 + (3A-8B)x - 24
...
So, the possible values of A and B are:
A = 5, B = 2
A = -20/3, B = -3/2

Let's check both cases:
Case 1: A = 5, B = 2
(Ax-8)(Bx+3) = (5x-8)(2x+3)
Expanding, we get:
10x^2 + 15x - 16x - 24
= 10x^2 - x - 24
which is the given expression.
Case 2: A = -20/3, B = -3/2
(Ax-8)(Bx+3) = (-20/3 x-8)(-3/2 x+3)
Expanding, we get:
20x^2 - 30x + 24x - 72
= 20x^2 - 6x - 72
which is not the given expression.
...
Therefore, AB + B = 12.
The final answer is \\boxed{12}.

Figure 21: A comparison of DeepSeek-Math-7B's "Enumeration" behavior before and after zero RL training. Here, "base solution" represents the response of the DeepSeek-Math-7B base model, while "zero solution" represents the response of the model after training.

**Question**

Let $f(x) = x^3 + 3x^2 + 1.$ There exist real numbers $a \neq 0$ and $b,$ such that
$$f(x) - f(a) = (x - a)^2 (x - b).$$ Enter the ordered pair $(a,b).$

**Base Solution**

Factor the left side of the equation as follows:
$$f(x) - f(a) = (x - a) (x^2 + 2x + 1 + 3x + 3a).$$

Compare the coefficients of $x^2$ and $x$ in the two expressions for $f(x) - f(a)$ to find the values of $a$ and $b$ that satisfy the equation. The solution is:
$$\boxed{(a, b) = (-1, -1)}$$

**Zero Solution**

To find the real numbers $a \neq 0$ and $b$ such that
$$f(x) - f(a) = (x - a)^2 (x - b),$$
we first need to compute $f(a)$:
$$f(a) = a^3 + 3a^2 + 1.$$
.......
Let's check if $b = 0$ is a valid solution:
Substituting $b = 0$ into $b + 2a = -3$, we get:
$$0 + 2a = -3 \implies 2a = -3 \implies a = -\frac{3}{2}.$$
Now, we can verify that this value of $a$ also satisfies
the equation $2ab + a^2 = 0$:
$$2ab + a^2 = 2 \left( -\frac{3}{2} \right) \cdot 0 + \left( -\frac{3}{2} \right)^2 = 0 + \frac{9}{4} = \frac{9}{4}.$$
However, this does not equal $0$. So, $b = 0$ is not a valid solution.
Let's try another possibility:
.....
Therefore, the ordered pair $(a, b)$ is $\boxed{(-2, 1)}$.

Verification

Backtracking

Figure 22: A comparison of Mistral-24B's "verification" and "backtraining" behavior before and after "zero training." Here, "base solution" represents the response of the Mistral-24B base model, while "zero solution" represents the response of the model after training.

**Question**

The mean, median, and unique mode of a list of positive integers are three consecutive integers in some order. Compute the least possible sum of the integers in the original list.

**Base Solution**

Step-by-step solution:<|im

**Zero Solution**

Let the list of positive integers be $\( L = \{x_1, x_2, \ldots, x_n\} \)$ where $\( n \)$ is the number of elements in the list.\nWe need to find the least possible sum of these integers given that the mean, median, and unique mode of $\( L \)$ are three consecutive integers.\nLet's denote the mean as $\( \mu \)$, the median as $\( M \)$, and the mode as $\( m \)$.

...

To minimize the sum of the elements in the list, we should find the smallest possible values for $\( \mu \)$, $\( M \)$, and $\( m \)$ that satisfy the conditions.

Let's start with the simplest case where the list has a small number of elements.

Case 1: $\( n = 1 \)$\nIn this case, the mean, median, and mode are all the same number, which contradicts the requirement that they are three consecutive integers.

...

Case 3: $\( n = 3 \)$\nLet the three elements in the list be $\( x_1 \)$, $\( x_2 \)$, and $\( x_3 \)$.\nMean $\( \mu = \frac{x_1 + x_2 + x_3}{3} \)$\nMedian $\( M = x_2 \)$ (since the elements are in ascending order)\nMode: Since $\( n = 3 \)$, the mode can be $\( x_1 \)$, $\( x_2 \)$, or $\( x_3 \)$. It's not guaranteed that the mode will be unique.\nLet's consider the possible values for $\( x_1 \)$, $\( x_2 \)$, and $\( x_3 \)$ such that $\( \mu \)$, $\( M \)$, and $\( m \)$ are three consecutive integers.\nWithout loss of generality, let's assume $\( x_1 \leq x_2 \leq x_3 \)$.

Case 3.1: $\( \mu = k \)$, $\( M = k+1 \)$, $\( m = k+2 \)$\n$\( M = x_2 = k+1 \)$\n$\( \mu = \frac{x_1 + x_2 + x_3}{3} = k \)$\n$\( x_1 + x_2 + x_3 = 3k \)$\n$\( x_1 + (k+1) + x_3 = 3k \)$\n$\( x_1 + x_3 = 3k - (k+1) \)$\n$\( x_1 + x_3 = 2k - 1 \)$\nMode $\( m = k+2 \)$\nSince the mode is unique, one of the values $\( x_1, x_2, x_3 \)$ must appear exactly once while the other two values appear the same number of times. Since $\( n = 3 \)$, each value can only appear at most once, so this case is not valid.

...

Since $\( n = 3 \)$, each value can only appear at most once, so this case is not valid.

Now, let's try to find the valid cases for $\( n = 3 \)$. For $\( n = 3 \)$, the three values can be:\n- $\( x_1 \)$, $\( x_2 \)$ and $\( x_3 \)$

Let's try the smallest possible values for $\( k \)$. Let $\( k = 1 \)$, then the three consecutive integers are 1, 2, and 3.

Case 3.1: $\( \mu = 1 \)$, $\( M = 2 \)$, $\( m = 3 \)$\n$\( \mu = \frac{x_1 + x_2 + x_3}{3} = 1 \)$\n$\( x_1 + x_2 + x_3 = 3 \)$

....

Now, we have found a possible set of values for $\( x_1 \)$, $\( x_2 \)$, $\( x_3 \)$, and $\( x_4 \)$ that satisfy the required conditions. Let's calculate the sum of the elements.\nThe sum of the elements in the list $\( \{1, 1, 3, 7\} \)$ is $\( 1 + 1 + 3 + 7 = 12 \)$.\nTherefore, the least possible sum of the integers in the original list is $\(\boxed{12}\)$.

Figure 23: A comparison of Mistral-Small-24B's "Enumeration" behavior before and after zero RL training. Here, "base solution" represents the response of the Mistral-Small-24B base model, while "zero solution" represents the response of the model after training.

**Question**

Compute $a+b+c,$ given that $a,$ $b,$ and $c$ are the roots of $$\frac{1}{x} + 5x^2 = 6x - 24.$$

**Base Solution**

To find the constant term $c$, what we want is that the polynomial\n$$\frac{1}{x} + 5x^2 - 92x + c = 0$$\nis the solution of the given polynomial\n$$\frac{1}{x} + 5x^2 = 6x - 24.$$\nNow to find $c$, we factor $g(x) = \frac{1}{x} + 5x^2 + 4x - 24 = \frac{5x^3 + 2x^2 - 26x - 24}{x} = 5x^2 + \frac{2x^2 - 26x - 24}{x}.$ So,\n$$\frac{5x^2}{x} + \frac{2x^2}{x} - \frac{26x}{x} - \frac{24}{x} = 6x - 24.$$\nSimplifying, we have $5x^2 + 2x^2 - 26x - 24 - 2x^2 = 0$\n$\implies x^4 - 30x - 24 = 0.$\nIf $g\_1,$ $g\_2,$ and $g\_3$ are the roots of $g\_1 = g\_2$ and $g\_1 = g\_3$, then we can conclude that $g\_1$ and $g\_2$ are roots of $g$.\nSo $g = g\_1 \cdot g\_2 = g\_3^2.$\nNow $6g - 24 + c = 0,$ so $c = 24 - 6g.$ We want the quotient, which is the form\n$$6\left(x^4 - 30x - 24\right) = 6x^4 - 180x - 144.$$

**Zero Solution**

To find the roots $a$, $b$, and $c$ of the equation $(\frac{1}{x} + 5x^2 = 6x - 24)$, we first need to rewrite the equation in a standard quadratic form:\n\n$$\frac{1}{x} + 5x^2 - 6x + 24 = 0.$$\n\nMultiplying through by $x$ to clear the fraction, we get:\n\n$$1 + 5x^3 - 6x^2 + 24x = 0.$$\n\nRearranging terms, we obtain the polynomial:\n\n$$5x^3 - 6x^2 + 24x + 1 = 0.$$\n\nWe need to solve this cubic equation for $x$. By the Rational Root Theorem, the possible rational roots are $(\pm 1, \pm \frac{1}{5})$.

Let's test these possible roots.
Testing $(x = -1)$:\n\n$$5(-1)^3 - 6(-1)^2 + 24(-1) + 1 = -5 - 6 - 24 + 1 = -34 \neq 0.$$

Testing $(x = \frac{1}{5})$:\n\n$$5\left(\frac{1}{5}\right)^3 - 6\left(\frac{1}{5}\right)^2 + 24\left(\frac{1}{5}\right) + 1 = 5 \cdot \frac{1}{125} - 6 \cdot \frac{1}{25} + \frac{24}{5} + 1 = \frac{5}{125} - \frac{6}{25} + \frac{24}{5} + 1 = \frac{1}{25} - \frac{6}{25} + \frac{120}{25} + \frac{25}{25} = \frac{1 - 6 + 120 + 25}{25} = \frac{140}{25} = 0.$$\n\nSo, $(x = \frac{1}{5})$ is a root. We can factor $(x - \frac{1}{5})$ out of the polynomial $(5x^3 - 6x^2 + 24x + 1)$. Using synthetic division:\n\n$$\begin{array}{r|rrrr}\n\frac{1}{5} & 5 & -6 & 24 & 1 \\\n   &   & 1 & -\frac{1}{5} & \frac{11}{25} \\\n\hline\n   & 5 & -5 & 23 & 0 \\\n\end{array}$$\n\nThe quotient is $(5x^2 - 5x + 1)$. So, we have:\n\n$$5x^3 - 6x^2 + 24x + 1 = (x - \frac{1}{5})(5x^2 - 5x + 1).$$\n\nNext, we solve the quadratic equation $(5x^2 - 5x + 1 = 0)$ using the quadratic formula $(x = \frac{-b \pm \sqrt{b^2 - 4ac}}{2a})$:\n\n$$x = \frac{5 \pm \sqrt{25 - 20}}{10} = \frac{5 \pm \sqrt{5}}{10}.$$\n\nSo, the roots are $(x = \frac{1}{5})$, $(x = \frac{5 + \sqrt{5}}{10})$, and $(x = \frac{5 - \sqrt{5}}{10})$. Therefore, the roots $a$, $b$, and $c$ are $(a = \frac{1}{5})$, $(b = \frac{5 + \sqrt{5}}{10})$, and $(c = \frac{5 - \sqrt{5}}{10})$.\nThe sum of the roots $(a + b + c)$ is:\n\n$$a + b + c = \frac{1}{5} + \frac{5 + \sqrt{5}}{10} + \frac{5 - \sqrt{5}}{10} = \frac{2}{10} + \frac{5 + \sqrt{5} + 5 - \sqrt{5}}{10} = \frac{2}{10} + \frac{10}{10} = \frac{2 + 10}{10} = \frac{12}{10} = \frac{6}{5}.$$\n\nThus, the value of $(a + b + c)$ is:\n\n$$\boxed{\frac{6}{5}}.$$

Figure 24: A comparison of Qwen-2.5-0.5B's "Verification" behavior before and after zero RL training. Here, "base solution" represents the response of the Qwen-2.5-0.5B base model, while "zero solution" represents the response of the model after training.

**Question**

The sum of the digits of a two-digit number is $13.$ The difference between the number and the number with its digits reversed is $27.$ What is the sum of the original number and the number with its digits reversed?

**Base Solution**

The tens digit can be any digit from 1 to 9. To get a two-digit number, the tens digit has to be 8 and the units digit can be 9. The sum of the tens digit and the units digit is 8 + 9 = 17. From the given information, the difference between the number with digits reversed and the original number is 27. If we know that the difference is 27, then the sum of the units digit and the tens digit will give us the difference. Hence, we can form the following equation 8 + 9 - (17 - 27) = 4 + 18 = 22. Therefore, the sum of the original number and the number with its digits reversed is 22.

**Zero Solution**

Let the two-digit number be \\(10a + b\\), where \\(a\\) and \\(b\\) are the digits. According to the problem, we have two equations:\n1. \\(a + b = 13\\)\n2. The difference between the number and the number with its digits reversed is 27, so \\(10b + a - (10a + b) = 27\\).\n\nSimplifying the second equation, we get:\n\\[10b + a - 10a - b = 27\\]\n\\[9b - 9a = 27\\]\n\\[b - a = 3\\]\n\nNow we have a system of linear equations:\n1. \\(a + b = 13\\)\n2. \\(b - a = 3\\)\n\nWe can solve these equations simultaneously. Adding the two equations, we get:\n\\[a + b + b - a = 13 + 3\\]\n\\[2b = 16\\]\n\\[b = 8\\]\n\nSubstituting \\(b = 8\\) into the first equation, we get:\n\\[a + 8 = 13\\]\n\\[a = 5\\]\n\nSo, the two-digit number is \\(10a + b = 10(5) + 8 = 58\\).\n\nThe number with its digits reversed is \\(85\\). The difference between the original number and the number with its digits reversed is:\n\\[58 - 85 = -27\\]

However, we are given that the difference is 27, which indicates there might be a misunderstanding in the problem statement or a need to re-evaluate the conditions. Let's recheck the conditions and solve again.

Given:\n1. \\(a + b = 13\\)\n2. \\(b - a = 3\\)\n\nWe can solve these equations again. Adding the two equations, we get:\n\\[a + b + b - a = 13 + 3\\]\n\\[2b = 16\\]\n\\[b = 8\\]\n\nSubstituting \\(b = 8\\) into the first equation, we get:\n\\[a + 8 = 13\\]\n\\[a = 5\\]\n\nSo, the two-digit number is \\(10a + b = 10(5) + 8 = 58\\).\n\nThe number with its digits reversed is \\(85\\). The difference between the original number and the number with its digits reversed is:\n\\[58 - 85 = -27\\]\n\nIt seems there was a misunderstanding. Let's re-evaluate the problem correctly. The correct interpretation should be:\n1. \\(a + b = 13\\)\n2. \\(b - a = 3\\)\n\nSolving these equations correctly, we get:\n\\[a = 5\\]\n\\[b = 8\\]\n\nSo, the two-digit number is \\(10a + b = 10(5) + 8 = 58\\).\n\nThe number with its digits reversed is \\(85\\). The difference between the original number and the number with its digits reversed is:\n\\[58 - 85 = -27\\]\n\nIt appears there was a miscalculation. Let's recheck the conditions again. The correct solution should be:\nThe sum of the original number and the number with its digits reversed is:\n\\[58 + 85 = 143\\]\n\nThus, the sum of the original number and the number with its digits reversed is:\n\\[\\boxed{143}\\]

Figure 25: A comparison of Qwen-2.5-0.5B's "Backtracking" behavior before and after zero RL training. Here, "base solution" represents the response of the Qwen-2.5-0.5B base model, while "zero solution" represents the response of the model after training.

**Question**

Compute the least integer greater than 2023 , the sum of whose digits is 17 .

**Base Solution**

First, note that, the sum of the digits of 2023 is 17 , so 2023 is 5 less than double of half of 2023. That is, 2023 is some integer n and 2023 - 5 = n/2 . 2023 is 5 less than double 2023. Since the difference 5 is odd, it follows that n must be odd, so that &strong;6 is the least nonnegative integer greater than or equal to 6+5=11. Therefore, the least integer greater than 2023, whose sum of digits is 17, is 11.\n####\nThe answer is n.

**Zero Solution**

To find the least integer greater than 2023 whose sum of digits is 17, we need to follow these steps:\n\n1. Calculate the sum of the digits of 2023.\n2. Find the smallest integer greater than 2023 whose sum of digits is 17.\n3. Determine the least integer greater than 2023 that meets the criteria.\n\nFirst, let's calculate the sum of the digits of 2023:\n\\[ 2 + 0 + 2 + 3 = 7 \\]\nSo, the sum of the digits of 2023 is 7. We need to find the smallest integer greater than 2023 whose sum of digits is 17.\n\nLet's denote the smallest integer greater than 2023 by \\( x \\). We need to find \\( x \\) such that:\n\\[ x - 2023 < 17 \\]\n\\[ x > 2023 + 17 \\]\n\\[ x > 2040 \\]\nThe smallest integer greater than 2040 is 2041. Now, we need to check if 2041 has a sum of digits equal to 17:\n\\[ 2 + 0 + 4 + 1 = 7 \\]

Since 2041 does not have a sum of digits equal to 17, we need to check the next integer, which is 2042:\n\\[ 2 + 0 + 4 + 2 = 8 \\]
Since 2042 does not have a sum of digits equal to 17, we continue checking the next integer, which is 2043:\n\\[ 2 + 0 + 4 + 3 = 9 \\]
Since 2043 does not have a sum of digits equal to 17, we continue checking the next integer, which is 2044:\n\\[ 2 + 0 + 4 + 4 = 10 \\]
Since 2044 does not have a sum of digits equal to 17, we continue checking the next integer, which is 2045:\n\\[ 2 + 0 + 4 + 5 = 11 \\]
Since 2045 does not have a sum of digits equal to 17, we continue checking the next integer, which is 2046:\n\\[ 2 + 0 + 4 + 6 = 12 \\]
Since 2046 does not have a sum of digits equal to 17, we continue checking the next
...
Since 2066 does not have a sum of digits equal to 17, we continue checking the next integer, which is 2067:\n\\[ 2 + 0 + 6 + 7 = 15 \\]
Since 2067 does not have a sum of digits equal to 17, we continue checking the next integer, which is 2068:\n\\[ 2 + 0 + 6 + 8 = 15 \\]
Since 2068 does not have a sum of digits equal to 17, we continue checking the next integer, which is 2069:\n\\[ 2 + 0 + 6 + 9 = 17 \\]
Since 2069 has a sum of digits equal to 17, the least integer greater than 2023 whose sum of digits is 17 is:\n\\[ \\boxed{2069} \\]

Figure 26: A comparison of Qwen-2.5-0.5B's "Enumeration" behavior before and after zero RL training. Here, "base solution" represents the response of the Qwen-2.5-0.5-B base model, while "zero solution" represents the response of the model after training.

**Question**

Let \\[f(n) =\n\\begin{cases}\n4n+3 &\\text{if }n<a, \\\\\n7n-12 &\\text{if }n\\ge{a}.\n\\end{cases}\n\\]Find $a$ if the graph of $y=f(n)$ is continuous.

**Base Solution**

To make the graph continuous, the values of $f(n)$ when $n<a$, and when $n\\ge{a}$, must be equal.\nSubstituting $n=a$ into $7n-12$, we find $7a-12$. Substituting $n$ into $4n+3$, we find $4a+3$. Setting these equal to each other and solving for $a$, we find $a=3$.\nThe answer is 3

**Zero Solution**

To determine the value of \\( a \\) that makes the function \\( f(n) \\) continuous, we need to ensure that the left-hand limit as \\( n \\) approaches \\( a \\) from the left equals the right-hand limit as \\( n \\) approaches \\( a \\) from the right, and both equal the value of the function at \\( n = a \\).\n\nThe function \\( f(n) \\) is defined as:\n\\[\nf(n) =\n\\begin{cases}\n4n+3 & \\text{if } n < a, \\\\\n7n-12 & \\text{if } n \\ge a.\n\\end{cases}\n\\]\n\nFor the function to be continuous at \\( n = a \\), the following must be true:\n\\[\n\\lim_{n \\to a^-} f(n) = \\lim_{n \\to a^+} f(n) = f(a)\n\\]\n\nFirst, let's find the left-hand limit as \\( n \\) approaches \\( a \\):\n\\[\n\\lim_{n \\to a^-} f(n) = 4a + 3\n\\]\n\nNext, let's find the right-hand limit as \\( n \\) approaches \\( a \\):\n\\[\n\\lim_{n \\to a^+} f(n) = 7a - 12\n\\]\n\nSince the function is continuous at \\( n = a \\), we set the left-hand limit equal to the right-hand limit:\n\\[\n4a + 3 = 7a - 12\n\\]\n\nNow, solve for \\( a \\):\n\\[\n4a + 3 = 7a - 12\n\\]\n\\[\n3 + 12 = 7a - 4a\n\\]\n\\[\n15 = 3a\n\\]\n\\[\na = 5\n\\]\n\nTo ensure that the function is continuous at \\( n = a \\),

we also need to check that the value of the function at \\( n = a \\) is the same from both sides. Substituting \\( a = 5 \\) into the function:\n\\[\nf(5) = 4(5) + 3 = 20 + 3 = 23\n\\]\n\nNow, let's verify the function values:

we also need to check that the value of the function at \\( n = a \\) is the same from both sides. Substituting \\( a = 5 \\) into the function:\n\\[\nf(5) = 4(5) + 3 = 20 + 3 = 23\n\\]\n\nNow, let's verify the function values:

- For \\( n < 5 \\), \\( f(n) = 4n + 3 \\). At \\( n = 5 \\):\n \\[\n f(5) = 4(5) + 3 = 20 + 3 = 23\n \\]\n- For \\( n \\ge 5 \\), \\( f(n) = 7n - 12 \\). At \\( n = 5 \\):\n \\[\n f(5) = 7(5) - 12 = 35 - 12 = 23\n \\]\n\nSince both expressions give \\( f(5) = 23 \\), the function is continuous at \\( n = 5 \\).\n\nTherefore, the value of \\( a \\) that makes the function \\( f(n) \\) continuous is \\(\\boxed{5}\\).

Figure 27: A comparison of Qwen-2.5-1.5B's "Verification" behavior before and after zero RL training. Here, "base solution" represents the response of the Qwen-2.5-1.5B base model, while "zero solution" represents the response of the model after training.

**Question**

Determine all linear functions $f(x)=ax+b$ such that if $g(x)=f^{-1}(x)$ for all values of $x$, then $f(x)-g(x)=44$ for all values of $x$. (Note: $f^{-1}$ is the inverse function of $f$.)

**Base Solution**

The first thought might be to work through the problem algebraically. Note that the question is really asking for a function with codomain of $-4\\------\\ 44$, in other words, for any $y$ between $40$ and $-40$, there should be a corresponding $x$ such that $f(x)=y$. But then, all we need is to reverse the variables, $f(g(x))-g(x)=44$.\nThis leads us to the idea of trying $f(x)=40+ax$ ($a$ time dilation, so that all such $a$'s cancel out), which leads\n$$f(x)+g(x)=44+2ax,$$\nwhich cannot be the untagged error $44$, $2a$ simply cannot be 1! This has no solution, meaning we iterate our kindest algorithmical thought.\nSo, back to the first trajectory, $g(x)=2-x$. So $f(g^{-1}(x))=f(2-x)=a(2-x)+b=2a-x+a+b=2a-x+3-x=40-x=2(x-40)=44$.\nSo, $x-40=22$, which means $x=62$.\nWe are done!\nLet's implement our solution in Python.

**Zero Solution**

To determine all linear functions \\( f(x) = ax + b \\) such that if \\( g(x) = f^{-1}(x) \\) for all values of \\( x \\), then \\( f(x) - g(x) = 44 \\) for all values of \\( x \\), we will follow these steps:\n\n1. **Find the inverse function \\( g(x) \\) of \\( f(x) \\):**\n\n   Given \\( f(x) = ax + b \\), we find the inverse function \\( g(x) \\) by solving for \\( x \\) in terms of \\( y \\) where \\( y = f(x) \\):\n   \\[\n   y = ax + b\n   \\]\n   Solving for \\( x \\):\n   \\[\n   y - b = ax \\implies x = \\frac{y - b}{a}\n   \\]\n   Therefore, the inverse function is:\n   \\[\n   g(x) = \\frac{x - b}{a}\n   \\]\n\n2. **Set up the given condition \\( f(x) - g(x) = 44 \\):**\n\n   We know that \\( f(x) = ax + b \\) and \\( g(x) = \\frac{x - b}{a} \\). Substitute these into the condition:\n   \\[\n   f(x) - g(x) = (ax + b) - \\frac{x - b}{a}\n   \\]\n   Simplify the right-hand side:\n   \\[\n   ax + b - \\frac{x - b}{a} = ax + b - \\frac{x}{a} + \\frac{b}{a}\n   \\]\n   Combine the terms:\n   \\[\n   ax - \\frac{x}{a} + b + \\frac{b}{a} = \\frac{a^2x}{a} - \\frac{x}{a} + b + \\frac{b}{a} = \\frac{a^2x - x}{a} + b + \\frac{b}{a}\n   = \\frac{(a^2 - 1)x}{a} + b + \\frac{b}{a}\n   \\]\n   Simplify further:\n   \\[\n   \\frac{(a^2 - 1)x}{a} + \\frac{ab + b}{a} = \\frac{(a^2 - 1)x + ab + b}{a}\n   \\]\n   According to the given condition, this equals 44 for all \\( x \\):\n   \\[\n   \\frac{(a^2 - 1)x + ab + b}{a} = 44\n   \\]\n   For this to hold for all \\( x \\), the coefficient of \\( x \\) must be zero (since the left-hand side is a linear function in \\( x \\)):\n   \\[\n   a^2 - 1 = 0 \\implies a^2 = 1 \\implies a = \\pm 1\n   \\]

3. **Consider the two cases for \\( a \\):**\n\n   - **Case 1: \\( a = 1 \\)**\n   \\[\n   f(x) = x + b\n   \\]\n   \\[\n   g(x) = \\frac{x - b}{1} = x - b\n   \\]\n   Substitute into the condition:\n   \\[\n   f(x) - g(x) = (x + b) - (x - b) = x + b - x + b = 2b\n   \\]\n   Set equal to 44:\n   \\[\n   2b = 44 \\implies b = 22\n   \\]\n   Thus, \\( f(x) = x + 22 \\).\n\n   - **Case 2: \\( a = -1 \\)**\n   \\[\n   f(x) = -x + b\n   \\]\n   \\[\n   g(x) = \\frac{x - b}{-1} = -x + b\n   \\]\n   Substitute into the condition:\n   \\[\n   f(x) - g(x) = (-x + b) - (-x + b) = -x + b + x - b = 0\n   \\]\n   This does not satisfy the condition \\( f(x) - g(x) = 44 \\).

4. **Conclusion:**\n\n   The only linear function \\( f(x) = ax + b \\) that satisfies the given condition is when \\( a = 1 \\) and \\( b = 22 \\). Therefore, the linear function is:\n   \\[\n   \\boxed{f(x) = x + 22}\n   \\]

Figure 28: A comparison of Qwen-2.5-1.5B's "Enumeration" behavior before and after zero RL training. Here, "base solution" represents the response of the Qwen-2.5-1.5B base model, while "zero solution" represents the response of the model after training.

**Question**

Find all values of $x$ that satisfy the equation $x = \\!\\sqrt{11-2x} + 4$.

**Base Solution**

$ x = \\!\\sqrt{11-2x} + 4$\nSubtract 4 on the both sides\n$x-4 = \\!\\sqrt{11-2x}$\nSquare on both sides$ (x-4)^2 = \\!\\sqrt{11-2x}$\nSimplify $(x-4)^2 = 11-2x$\nExpand $(x^4 - 3x + 12) = 11-2x$\n$5x^2 - x=13$ Wait. Rewriting that square is not right: $x-4 = 11-2x$ (x-4 + 2x = 11)\n$ 4x + 12$\n$ -3x^2 + x +2x -4x = 13$\n$-3x^2 + -2x =13$\n$-3x^2 -2x - 13 = 0$\nB is our case : $(-b / 2a)^2..$ \nHere's what we have $b = -2$, and $-3 / -2$ equals 1.5. Now we had to find the square. So the square is 2.25. But what about $-3x^2 + -2x =13$? Since our coefficient is -3, we multiply it after we take away the -2. 2.25 X (-3) = -6.75. And +13 equals 6.25. I don't know if my methods are correct. I need a one-line solution or a calculus equation so I can solve it.

**Zero Solution**

Given: $x = \\!\\sqrt{11-2x} + 4$\nSubtract 4 from both sides, we get\n$x - 4 = \\!\\sqrt{11-2x}$\nSquare both sides, we get\n$(x - 4)^2 = 11-2x$\n$x^2 - 8x + 16 = 11-2x$\n$x^2 - 10x + 5 = 0$\n$x = \\frac{10 \\pm \\sqrt{100 - 20}}{2} = \\frac{10 \\pm \\sqrt{80}}{2} = \\frac{10 \\pm 4\\sqrt{5}}{2} = 5 \\pm 2\\sqrt{5}$\n$x = 5 + 2\\sqrt{5}$ or $x = 5 - 2\\sqrt{5}$\n$x = 5 + 2\\sqrt{5}$ satisfies the equation.\n$x = 5 + 2\\sqrt{5} = \\!\\sqrt{11-2(5 + 2\\sqrt{5})} + 4$\n $= \\!\\sqrt{11 - 10 - 4\\sqrt{5}} + 4$\n $= \\!\\sqrt{1 - 4\\sqrt{5}} + 4$\n $= \\!\\sqrt{1 - 4\\sqrt{5}} + 4$

Since, $\\sqrt{1 - 4\\sqrt{5}}$ is not a real number, $x = 5 + 2\\sqrt{5}$ is not a solution.

$x = 5 - 2\\sqrt{5} = \\!\\sqrt{11-2(5 - 2\\sqrt{5})} + 4$\n $= \\!\\sqrt{11 - 10 + 4\\sqrt{5}} + 4$\n $= \\!\\sqrt{1 + 4\\sqrt{5}} + 4$\n $= \\!\\sqrt{1 + 4\\sqrt{5}} + 4$\n$\\!\\sqrt{1 + 4\\sqrt{5}}$ is a real number. Hence, $x = 5 - 2\\sqrt{5}$ is a solution.\n$x = 5 - 2\\sqrt{5}$

Figure 29: A comparison of Llama-3.1-8B's "Verification" behavior before and after zero RL training. Here, "base solution" represents the response of the Llama-3.1-8B base model, while "zero solution" represents the response of the model after training.

**Question**

The sum of the digits of a two-digit number is $13.$ The difference between the number and the number with its digits reversed is $27.$ What is the sum of the original number and the number with its digits reversed?

**Base Solution**

1) In a two digit number. $$L \cdot Tens + R \cdot Ones$$ with,\n$$L \in [\\!
[0,9]\\!]$$ and\n$$R \in [\\![1,9]\\!]$$\n2) If sum of the digits is $$13$$, then (hypothesis).\n$$\\Leftrightarrow L+R=13$$\n3) By the hypothesis, number $N$ and it's reversed number $N'$\nare such that $$N-N'=27$$\nand we have $$L \cdot Tens + R \cdot Ones = L \cdot Ones + R \cdot Tens$$ So, $$N-N' = (L-R)(Ten-Ones)-(L-R)(One-Tens)=180$$ which is equivalent to $$55$$. The sum $N+N'$ of these.\nWe have $$N+N' =(L+R) \cdot Ones + (L+R) \cdot Tens$$ $$+$$ $$+(L+R) \cdot Ones + (L+R) \cdot Tens$$ So, $$2 \cdot (L+R) \cdot Ones$$ $$+$$ $$+ 2 \cdot (L+R) \cdot Tens $$ which is equal to $220=2 \cdot 110=2 \cdot (10 \cdot 11) \quad.$

**Zero Solution**

The sum of the digits of a two-digit number is $13$
Let digit at unit's place = a\nand digit at ten's place = b\na + b = 13
Next, the difference between the number and the number with its digits reversed is
$27.$
Number = 10*b + a = 10*b + a
Number with digits reversed = 10*a + b
Difference between the number and the number with its digits reversed = 10*b + a - (10*a
+ b) = 10*b + a - 10*a - b = 10*b - 10*a + a - b = 9*b - 9*a = 27
9*b - 9*a = 279*b = 9*a + 27\nb = a + 3
Substituting b = a + 3 in a + b = 13, we get\na + (a + 3) = 13\n2a + 3 = 13\n2a = 10\na
= 5\nb = a + 3 = 5 + 3 = 8
Number = 10*b + a = 10 * 8 + 5 = 80 + 5 = 85
Number with digits reversed = 10*a + b = 10 * 5 + 8 = 50 + 8 = 58
Difference between the number and the number with its digits reversed = 85 - 58 = 27
Given, Difference between the number and the number with its digits reversed = 27
So, 85 - 58 = 27
Thus, 85 + 58 = 143

Figure 30: A comparison of Llama-3.1-8B's "Subgoal Setting" behavior before and after zero RL training. Here, "base solution" represents the response of the Llama-3.1-8B base model, while "zero solution" represents the response of the model after training.

