# OpenReview forum: "SimpleRL-Zoo: Investigating and Taming Zero Reinforcement Learning for Open Base Models in the Wild"
_colmweb.org/COLM/2025/Conference — COLM 2025_

### Official Review · Reviewer_7p56 · 2025-05-06

**Rating:** 7
**Confidence:** 3
**Ethics Flag:** 1

**Summary:**

This work introduces SimpleRL‑Zoo, a zero‑shot reinforcement‑learning recipe that fine‑tunes ten open‑source LLMs. The approach combines a binary correctness reward with a length‑rectified GRPO objective and trains on GSM8K and MATH tasks divided by difficulty. Throughout training it records auxiliary diagnostics – Clip Ratio, Average Stopped Length, and a GPT‑4o‑based reasoning‑behavior probe – to characterise exploration and chain‑of‑thought dynamics. Additional experiments examine how reward format, curriculum difficulty and prior SFT influence training behaviour across the model zoo.

**Questions To Authors:**

- **Quantitative behaviors:** Could you provide any overall quantitative analysis for each reasoning behavior together with human vs. GPT‑4o agreement scores?
- **Domain transferability:** Have you tried SimpleRL-Zoo on non‑math datasets such as planning tasks, and do the obtained insights still apply?
- **Reward‑scale interaction:** Do you have results for Mistral-Small-24B or Llama3.1-70B under varying format‑reward strengths to establish the threshold at which strictness harms training?

**Reasons To Accept:**

- **Extensive empirical coverage:** A single minimalist recipe is validated on ten base models across five parameter scales and more than ten downstream benchmarks, providing strong evidence of generality. This diversity in model scale and task type enhances the credibility of the approach. The consistency across both small and large models highlights the robustness of the proposed method.
- **Plausible, domain‑aligned metric suite:** The paper justifies and formalises Clip Ratio, Average Stopped Length and a reasoning‑behavior probe, offering interpretable signals for length collapse, exploration and reasoning depth. These metrics provide nuanced insight into how model behavior evolves under RL training. They offer practical signals that help interpret training dynamics more clearly.
- **Actionable training guidelines:** Ablations provide concrete recommendations(e.g., avoid strict format rewards, align difficulty with base proficiency) that practitioners can adopt immediately for stable zero RL. The experiments also reveal which configurations lead to collapse, making the findings directly useful in real-world settings. The findings provide useful heuristics for stabilizing training across setups.

**Reasons To Reject:**

- **Lack of quantitative behavior validation:** The GPT‑4o‑labelled reasoning behaviors are plotted qualitatively but absolute ratios, error bars and human‑agreement statistics are omitted, limiting confidence in the analysis. Without such validation, it is difficult to assess whether the trends reflect real behavioral changes.This makes it harder to assess the reliability of the claimed behavior trends.
- **Math‑centric experimental scope:** Both training data and the majority of evaluations are restricted to mathematical reasoning. This limits the ability to generalize findings to broader tasks like planning or code generation. The insights may not directly transfer beyond math-focused benchmarks.
- **Unresolved reward‑strength threshold:** Section 3.1 tests strict vs. loose format rewards only on 7-8B models; without a sweep over larger models or intermediate penalty strengths, the work cannot pinpoint how strong a format penalty must be before performance degrades, nor whether the sensitivity shifts with scale. The current evaluation leaves open how model size interacts with reward formulation.

---

> ### Author Response · Authors · 2025-06-03
> **Response to Reviewer 7p56**
>
> **Q1: Lack of quantitative behavior validation？**
>
> **A1**: Thank you for your suggestion. We assess the consistency between GPT-4o-labeled reasoning behaviors and human annotations by having human experts annotate 105 model outputs. The table below presents the prediction rates and agreement rate. The prediction rate reflects how frequently each reasoning behavior is identified, while the agreement rate is the proportion of data on which the labelers (Human and GPT-4o) make the same prediction.
>
> Our results indicate a generally good level of agreement between GPT-4o and human annotations. However, GPT-4o tends to be more conservative when labeling certain behaviors such as Verification and Subgoal Setting. Upon closer examination, we observe that in cases with long CoT containing multiple reasoning behaviors, the model often favors labeling more obvious behaviors like Enumeration, while overlooking subtler ones.
>
> | Behavior        | Score by GPT-4o (%) | Score by Human (%) | Raw Agreement (%) |
> |----------------|---------------------|---------------------|--------------------|
> | Verification    | 78.10% (82/105)     | 85.71% (90/105)     | 90.48% (95/105)    |
> | Backtracking    | 33.33% (35/105)     | 35.24% (37/105)     | 98.10% (103/105)   |
> | Subgoal Setting | 66.67% (70/105)     | 74.29% (78/105)     | 90.48% (95/105)    |
> | Enumeration     | 61.90% (65/105)     | 63.81% (67/105)     | 94.29% (99/105)    |
>
> **Q2: Math‑centric experimental scope?**
>
> **A2**: Thank you for your suggestion. We conduct experiments in the science domain during the rebuttal phrase and observe changes in evaluation metrics, including accuracy, output length, clip ratio, and reasoning behavior — as detailed in General Response Q1. It shows zero RL training improves performance on science reasoning tasks and enhances critical reasoning behaviors such as subgoal setting, verification, and enumeration. Additionally, we verify the effect of the format reward discussed in Section 3, with supporting results provided in General Response Q2.  Our findings support the conclusion that overly strict format penalties can restrict model exploration and ultimately reduce overall performance.
>
> Furthermore, for the model trained on math reasoning in our paper, we also evaluated its performance on additional benchmarks: IFEVAL (instruction following), MMLU (general knowledge), and GPQA-Diamond (domain-specific expertise in chemistry, physics, and biology), as reported in Lines 145–147 of the main paper and Table 2 in Appendix E. The results demonstrate that zero RL training on math reasoning tasks not only improves performance in math reasoning itself, but also leads to notable gains on a broad range of tasks across different domains.
>
> **Q3: without a sweep over larger models or intermediate penalty strengths, the work cannot pinpoint how strong a format penalty must be before performance degrades, nor whether the sensitivity shifts with scale?**
>
> **A3**: Thank you for your suggestion. To experiment with larger models, we investigate the impact of format reward on training using Mistral-24B and find that even for this larger model, applying format reward significantly decreases performance. As shown in  Figure 1 below, introducing format reward leads to a reduction of approximately 4 points in peak performance and causes noticeable instability during training, with performance exhibiting sharp fluctuations. This conclusion is consistent with our finding in Section 3.1, supporting the conclusion that overly strict format penalties can hinder model exploration and ultimately degrade performance.
>
> For the point of sweeping over intermediate penalty strengths, this is a great suggestion. However, due to computational constraints and time limit of the rebuttal phase, we were unable to run multiple RL experiments with varying penalty strengths on a large model. We plan to explore this aspect and will discuss it in the next revision of the paper.
>
>
> Figure 1 (Mistral-24 B with/wo format reward)
>
> https://drive.google.com/file/d/1id9Wff-9n-O6VStlKVTnimKqfTpMkt3V/view?usp=sharing

---

> > ### Comment · Reviewer_7p56 · 2025-06-04
> >
> > Having comprehensively reviewed the authors’ experiments, I confirm that the authors have appropriately addressed the initial concerns and inquiries. Accordingly, I have increased the score.

---

> > > ### Author Response · Authors · 2025-06-04
> > >
> > > Thank you for your review and valuable suggestions！

---

### Official Review · Reviewer_ccrE · 2025-05-07

**Rating:** 8
**Confidence:** 2
**Ethics Flag:** 1

**Summary:**

Significance:
This paper helps the community understand effective setups and techniques for "zero" reasoning training in models.
Given that for-profit labs (e.g. OpenAI, Google) do not release their training techniques, it is helpful that we have papers such as these that list the important setup details that matter for reasoning training.
This is especially important since it appears that reasoning training is the near-term solution for increased SOTA performance on several benchmarks.
If we do not have science on what exactly is important, and what happens in the reasoning training process, the open scientific community will be unable to make progress in understanding these SOTA models.

Clarity:
I think the paper presents its findings clearly, using understandable graphs to show improvements in accuracy over training iterations.
One area of improvement: include CoT examples in the main text rather than the appendix. This would help the reader understand the results better e.g. what is "backtracking" and "verification". For seasoned readers, this may be obvious, but I think it would help a more general audience. The audience would also find it interesting. E.g. the "Aha" moment in the DeepSeek paper's main text was something that stood out to a general audience, and was easy to understand.

Originality and Quality:
The topic of exploring reasoning training is not novel. Several groups are exploring this space, as the paper mentions.
Still, I think the paper stands out through its good execution and detailed analysis. The authors provide concrete insights into various training specifics, I think is more important than the novelty of the idea itself.
We are at the early stage of understanding reasoning training, so work in this area is valuable, even if others are doing similar work. They often find different results (as the paper mentions). Having more research helps us converge on the best practices.

Overall, I think this paper is valuable. Still, I am less of an expert in the reasoning training of these models. It is difficult for me to verify if e.g. the exact training setup, and takeaways are reasonable compared to the related work.  so I will mark my review with lower confidence.

**Questions To Authors:**

Are the CoTs of these models mostly readable by humans? How often do we see code-switching in the CoTs? (Do we need to apply a language-switching loss, like DeepSeek did?) Are there special tokens that are not human-understandable, but the model uses to perform reasoning?
In the appendix, the CoTs look mostly readable. Perhaps, this is because we are examining in the particular the domain of math, and so I am curious about the CoTs when other domains are examined.

**Reasons To Accept:**

1. Provides valuable insights into training reasoning models. It studies a variety of open models (e.g. Mistral, Llama, Qwen, Mistral). Some of these have not been published previously e.g. Mistral.
- These are Interesting insights on the emergent properties of reasoning training. I found the fact that mistral had emergent reflective behavior (0 -> 50%) to be very interesting. This stands in contrast to some other models e.g. the Qwen models, that have minimal changes in behavior.
- Another example: The fact that SFT may harm exploration. This suggests we need to strike a balance between making the model be SFTed enough to follow instructions as a good prior (e.g. following a certain format), but not so much that it hurts exploration.

2. Provides reproducibility with open-source plans.
- The training setup and details will be helpful for future researchers in investigating emergent reasoning behavior.
- Having small open-source models trained on "zero" reasoning training is valuable for the community. For example, researchers studying chain-of-thought reasoning may want to inspect the CoT outputs of these models, and observe their properties that emerge.

**Reasons To Reject:**

One limitation is the sole focus on the domain of math, which may not be representative of what happens when we train these models on other domains (coding, science).
There is a question of whether this reproduces with larger models. But I concede that not everyone may have the compute needed to study  larger models.

---

> ### Author Response · Authors · 2025-06-03
> **Response to Reviewer ccrE**
>
> **Q1: One limitation is the sole focus on the domain of math？**
>
> **A1**: Thank you very much for your suggestions! We use zero RL to train models in the science domain during the rebuttal phrase and observe changes in accuracy, length, clip ratio, and reasoning behavior - please refer to General Response Q1. It shows zero RL training improves performance on science reasoning tasks and enhances critical reasoning behaviors such as subgoal setting, verification, and enumeration. We also verify the impact of having vs. not having the format reward discussed in Section 3 - please refer to General Response Q2. Our findings support the conclusion that overly strict format penalties can restrict model exploration and ultimately reduce overall performance.
>
> **Q2: Whether this reproduces with larger models? But I concede that not everyone may have the compute needed to study larger models.**
>
> **A2**: Thank you for your understanding. We already experimented with a 32B model, which, to the best of our knowledge, is the largest model used in most open-source training. Due to resource constraints, we were unable to conduct additional experiments with larger models during the rebuttal phase. However, we will definitely work with larger models if more resources become available in the future.
>
> **Q3: Are the CoTs of these models mostly readable by humans? Are there special tokens that are not human-understandable, but the model uses to perform reasoning? How often do we see code-switching in the CoTs?**
>
> **A3**: Thank you for your suggestion. We randomly selected 100 model outputs generated during training on math reasoning and science tasks and asked human experts to evaluate them. The CoTs in all outputs were readable and understandable. We found no special tokens or content that were unintelligible to humans, and none of the samples showed signs of code-switching. Furthermore, most open-source RL projects have not reported code-switching, suggesting that reinforcement learning on smaller open models is unlikely to induce code-switching or unreadable outputs.

---

> > ### Comment · Reviewer_ccrE · 2025-06-04
> >
> > Thank you for this analysis. Overall, I think this is a paper that is a clear accept and raised my score.

---

> > > ### Author Response · Authors · 2025-06-04
> > >
> > > Thank you very much for recognizing our work, and thank you for your suggestions and time.

---

### Official Review · Reviewer_S5Rk · 2025-05-07

**Rating:** 6
**Confidence:** 3
**Ethics Flag:** 1

**Summary:**

This paper applies the DeepSeek Zero Reinforcement Learning strategy to a number of smaller LLMs and demonstrates its viability.

**Questions To Authors:**

My first question to the authors is related to what I've described in the reasons to reject.

Second, related, question: is there any possibility to prove that the zero RL for one task helps in another task? This would be extraordinary.

**Reasons To Accept:**

The paper proves its "theory", that is, the zero RL training can be applied to smaller LLMs.
Indeed:
* Experiments are solid
* Results are convincing

**Reasons To Reject:**

My major concern is not with the paper itself, which only follows a trend, but with the trend itself. The idea of zero RL is that this way of learning can be applied **only** with tasks that are based on known algorithms and are tested on the same algorithms. Hence, it is not true that zero RL does not need training data. Indeed, it needs training data. These training data are the provided by the algorithms that generate responses given a problem.

Moreover, the paper does not help the reader to understand what is *zero RL*. This is taken as a standard technique also if this is emerged recently.

There is no analysis on how transformers learn these algorithmic ability of these algorithmic tasks. Hence, the paper does not help to dive deep into the architectures to explain the reason why this aha moment happens.

Beside the application of the technique to smaller models, there is no innovation, e.g., the extension of a zero RL technique to something that is not algorithmic.

---

> ### Author Response · Authors · 2025-06-03
> **Response to Reviewer S5Rk**
>
> **Q1: Concern about the trend of zero RL？"it is not true that zero RL does not need training data. Indeed, it needs training data"**
>
> **A1**: Thank you for your comment. We would like to clarify that zero RL never means zero training data, and we have never claimed in the paper that zero RL requires no training data. As in many other works, we use the name “zero RL” following DeepSeek-R1[1], where “zero” means "without any supervised fine-tuning" (also explained in Appendix A of our paper). While more precise naming of this paradigm is debatable, it is out of the scope of our paper.
>
> **Q2: The paper does not help the reader to understand what is zero RL. This is taken as a standard technique also if this is emerged recently.**
>
> **A2**: Thank you for the suggestion! We will provide additional context to offer more background on zero RL in the next revision of the paper.
>
> **Q3: Beside the application of the technique to smaller models, there is no innovation**
>
> **A3**: We respectfully disagree with this point. We emphasize that successfully applying the zero RL technique to smaller models is non-trivial, and our work makes both methodological and empirical contributions:
>
> - To the best of our knowledge, we are the first to demonstrate that the zero RL technique can effectively improve the reasoning performance of a diverse set of smaller and weaker models. Importantly, this process is not trivial; we propose key techniques—such as removing the format reward and controlling data difficulty—that are essential for the success.
>
> - We show that tracking response length alone is superficial and potentially misleading. To provide a more comprehensive assessment, we propose monitoring both the clip ratio and the average stopping length. By analyzing these cognitive behaviors, we also reveal that some models do not experience an "aha moment."
>
> - We revisit the traditional SFT+RL pipeline and find that standard SFT may actually diminish the gains achievable through RL.
>
> We note that most of these findings are novel and have not been previously reported in the literature.
>
> **Q4: The paper does not help to dive deep into the architectures to explain the reason why this aha moment happens?**
>
> **A4**: Zero RL for reasoning is still a rapidly evolving area of research. As mentioned in our Q3 response, this paper focuses on understanding how zero RL interacts with different base models and identifying the key factors that determine its success. Explaining aha moment from the architecture perspective is beyond the scope of our paper.
>
>
> **Q5: Is there any possibility to prove that the zero RL for one task helps in another task?**
>
> **A5**: We have already demonstrated this in Lines 145–147 of the main paper and in Table 2 of Appendix E, where we evaluated the model on other benchmarks: IFEVAL (instruction-following), MMLU (general knowledge), and GPQA-Diamond (domain-specific expertise in chemistry, physics, and biology). The results show that zero RL training on math reasoning tasks not only improves performance on those in-domain tasks but also enhances the model’s abilities across these diverse tasks. Other works like logic-RL [2] also demonstrate that using logic problems for zero RL training can generalize to challenging math benchmarks such as AIME and AMC.
>
> [1] Guo D, Yang D, Zhang H, et al. Deepseek-r1: Incentivizing reasoning capability in llms via reinforcement learning[J]. arXiv preprint arXiv:2501.12948, 2025.
>
> [2] Xie T, Gao Z, Ren Q, et al. Logic-rl: Unleashing llm reasoning with rule-based reinforcement learning[J]. arXiv preprint arXiv:2502.14768, 2025.

---

> > ### Comment · Reviewer_S5Rk · 2025-06-03
> > **Thank you for clarifying with A5**
> >
> > Comment **A5**: Given that part of the answer was in the appendix, I've overlooked this information.Thanks for clarifying. Please, move some of the content of the appendix in the main text of the paper.
> > Due to this part, I've consequently increased my score.

---

> > > ### Author Response · Authors · 2025-06-04
> > >
> > > Thank you very much for your time. If you have any other concerns, please feel free to raise them at any time. We are always willing to help resolve them for you.

---

### Official Review · Reviewer_9Rvw · 2025-05-22

**Rating:** 8
**Confidence:** 3
**Ethics Flag:** 1

**Summary:**

They carry out thorough experiments to understand how different attributes effect RL-zero performance/behaviors using a series of open LLMs on a number of different math benchmarks. They find that increasing response length during trining does not always correspond to the emergence of new cognitive behaviors. RL-zero works well on other open LLMs besides Qwen. Format rewards hurt exploration. It is best to have problems that are of the right difficulty for a given model. And SFT can hurt the performance of RL optimized models.

**Reasons To Accept:**

I think this paper presents overall very strong empirical work and many of their findings will likely be very useful for practitioners who want to train their own models with RL. I will point out that their experiments are very thorough, consisting of many different models, and evaluations. In such a fast moving field, empirical work like this which aims to understand existing techniques is very important.

**Reasons To Reject:**

I have no major reasons to reject that are worth noting.

---

> ### Author Response · Authors · 2025-06-03
> **Response to Reviewer 9Rvw**
>
> Thank you for recognizing our work！

---

### Author Response · Authors · 2025-06-03
**General Response**

**Q1: Applying zero RL training to other domains beyond math**

**A1**: To clearly demonstrate that zero RL training is effective in domains beyond mathematical reasoning, we conduct experiments using WebInstruct-verified [1] during the rebuttal phrase, covering diverse scientific fields such as Physics, Chemistry, Business, and Finance. Specifically, we train the Qwen2.5-7b models and evaluate their performance on the GPQA benchmark—a challenging dataset designed to measure domain-specific expertise in Chemistry, Physics, and Biology.

Figures 1,2,3 illustrate the impact of zero RL training on Qwen2.5-7b, highlighting metrics including Evaluation Accuracy, Response Length, Clip Ratio, Average Stopped Length, and reasoning behavior during evaluation. Our results confirm that zero RL training significantly enhances model performance in science-related reasoning tasks. Furthermore, we observe notable improvements in specific reasoning behaviors such as subgoal setting, verification, and enumeration, all of which contribute to an increased effective length of CoT.


Figure 1 (Accuracy&Length): (All links are anonymous)

https://drive.google.com/file/d/1XMl8bog6s33TRLEBb6BIlVlp3cNvFiV_/view?usp=sharing

Figure 2 (Clip Ratio&Average Stopped Length):

https://drive.google.com/file/d/1pMRAUcQbvbhFtaFESVq7WwBhVLQKZ2EL/view?usp=sharing

Figure 3 (Reasoning Behavior Ratio)

https://drive.google.com/file/d/1vz9muZZX3quo3Oa65GNo4doZzEzHKjzH/view?usp=sharing

**Q2: Can the findings in the paper be applied to other domains?**

**A2**: Building on Q1, we aim to verify some of the conclusions drawn in Section 3, particularly regarding the impact of format rewards. Given that WebInstruct-verified does not include difficulty labels, we did not conduct experiments on data difficulty.

Figure 4 below illustrates the changes in evaluation accuracy for Qwen2.5-7B, comparing runs with and without the use of format rewards. Notably, the model without format rewards achieved a peak accuracy approximately 2 points higher than the model with format rewards. Additionally, the accuracy improved more rapidly in the early training stages when format rewards were not applied. These observations align with the findings in Section 3.1, supporting the conclusion that overly strict format penalties can hinder model exploration and ultimately degrade performance.

Figure 4 (Qwen2.5-7B with/wo format reward):

https://drive.google.com/file/d/1gPkBKwODV42bkan5-_U3WOnSiPyIc6xk/view?usp=sharing



[1] Ma X, Liu Q, Jiang D, et al. General-reasoner: Advancing llm reasoning across all domains[J]. arXiv preprint arXiv:2505.14652, 2025.

---

### Decision · Program_Chairs · 2025-07-08

**Decision:**

Accept

**Comment:**

This paper offers a timely and well-executed empirical study of zero RL across ten diverse open-source language models. The authors provide practical insights such as avoiding strict format rewards, tuning task difficulty and understanding when SFT can hinder performance. Their analysis goes beyond surface metrics using novel diagnostics like Clip Ratio and reasoning-behavior probes to track emergent behaviors, including the "aha" moment even in smaller models.  The rebuttal further strengthens the work with new experiments in science domains and validation of behavior labels. This is a timely, high-impact contribution with clear value to the community. I recommend acceptance.